# Deterministic Policies for Constrained Reinforcement Learning in Polynomial Time

**Jeremy McMahan**
University of Wisconsin-Madison
jmcmahan@wisc.edu

## Abstract

We present a novel algorithm that efficiently computes near-optimal deterministic policies for constrained reinforcement learning (CRL) problems. Our approach combines three key ideas: (1) value-demand augmentation, (2) action-space approximate dynamic programming, and (3) time-space rounding. Our algorithm constitutes a fully polynomial-time approximation scheme (FPTAS) for any time-space recursive (TSR) cost criteria. A TSR criteria requires the cost of a policy to be computable recursively over both time and (state) space, which includes classical expectation, almost sure, and anytime constraints. Our work answers three open questions spanning two long-standing lines of research: polynomial-time approximability is possible for 1) anytime-constrained policies, 2) almost-sure-constrained policies, and 3) deterministic expectation-constrained policies.

## 1 Introduction

Constrained Reinforcement Learning (CRL) traditionally produces stochastic, expectation-constrained policies that can behave undesirably - imagine a self-driving car that randomly changes lanes or runs out of fuel. However, artificial decision-making systems must be predictable, trustworthy, and robust. One approach to ensuring these qualities is to focus on deterministic policies, which are inherently predictable, easily implemented [19], reliable for autonomous vehicles [30, 23], and effective for multi-agent coordination [38]. Similarly, almost sure and anytime constraints [36] provide inherent trustworthiness and robustness, essential for applications in medicine [15, 37, 32], disaster relief [18, 50, 45], and resource management [35, 34, 40, 7]. Despite the advantages of deterministic policies and stricter constraints, even the computation of approximate solutions has remained an open challenge since NP-hardness was proven nearly 25 years ago [19]. Our work addresses this challenge by studying the computational complexity of computing deterministic policies for general constraint criteria.

Consider a constrained Markov Decision Process (cMDP) denoted by $M$. Let $C$ represent an arbitrary cost criterion and $B$ be the available budget. We focus on the set of deterministic policies denoted by $\Pi^D$. Our objective is to compute: $\max_{\pi \in \Pi^D} V_M^\pi$ s.t. $C_M^\pi \leq B$, where $V_M^\pi$ is the value and $C_M^\pi$ is the cost of $\pi$ in $M$. This objective generalizes the example of a self-driving car calculating the fastest fixed route without running out of fuel. Our main question is the following:

> *Can near-optimal deterministic policies for constrained reinforcement learning problems be computed in polynomial time?*

Although optimal stochastic policies for expectation-constrained problems are efficiently computable [3], the situation drastically changes when we require deterministic policies and general constraints. Computing optimal deterministic policies is NP-hard for most popular constraints, including expectation [19], chance [51], almost sure, and anytime constraints [36]. This complexity remains even if we relax our goal to finding just one feasible policy, provided that we are dealing

with a single chance constraint [51], or at least two of the other mentioned constraints [36]. Beyond these computational challenges, traditional solution methods, such as backward induction [41, 3], fail to apply due to the cyclic dependencies among subproblems: the value of any decision may depend on the costs of both preceding and concurrent decisions, preventing a solution from being computed in a single backward pass.

**Past work.** Past approaches fail to simultaneously achieve computational efficiency, feasibility, and optimality. Optimal and feasible algorithms, albeit inefficient, utilize Mixed-Integer Linear Programs [17] and Dual-guided heuristic forward searches [29] for expectation-constraints, and cost-augmented MDPs for almost sure [11] and anytime constraints [36]. Conversely, optimal and efficient, though infeasible, algorithms are known for expectation [43], almost sure, and anytime constraints [36]. A fully polynomial-time approximation scheme (FPTAS) [49] is known for expectation constraints, but it requires strong assumptions such as a constant horizon [31]. Balancing computational efficiency, feasibility, and optimality remains the bottleneck to efficient approximation.

**Our contributions.** We present an FPTAS for computing deterministic policies under any time-space recursive (TSR) constraint criteria. A TSR criteria requires the cost of a policy to be computable recursively in both time and (state) space, which captures expectation, almost sure, and anytime constraints. Since non-TSR criteria, such as chance constraints [51], are provably inapproximable, TSR seems pivotal for efficient computation. Overall, our general framework answers three open complexity questions spanning two longstanding lines of work: we prove polynomial-time approximability for 1) anytime-constrained policies, 2) almost-sure-constrained policies, and 3) deterministic expectation-constrained policies, which have been open for nearly 25 years [19].

Our approach breaks down into three main ideas: (1) value-demand augmentation, (2) action-space approximate dynamic programming, and (3) time-space rounding. We augment the states with value demands and the actions with future value demands to break cyclic subproblem dependencies, enabling dynamic programming methods. Importantly, we use values because they can be rounded without compromising feasibility [36] and can capture constraints that are not predictable from cumulative costs. However, this results in an exponential action space that makes solving the Bellman operator as hard as the knapsack problem. By exploiting the space-recursive nature of the criterion, we can efficiently approximate the Bellman operator with dynamic programming. Finally, rounding value demands result in approximation errors over both time and space, but carefully controlling these errors ensures provable guarantees.

## 1.1 Related work

**Approximate packing.** Many stochastic packing problems, which generalize the knapsack problem, are captured by our problem. Dean et al. [16], Frieze and Clarke [21] derived optimal approximation ratio algorithms for stochastic packing and integer packing with multiple constraints, respectively. Yang et al. [52], Bhalgat et al. [6] designed efficient approximation algorithms for variations of the stochastic knapsack problem. Then, Halman et al. [27] derived an FPTAS for a general class of stochastic dynamic programs, which was then further improved in [26, 1]. These methods require a single-dimensional state space that captures the constraint. In contrast, our problems have an innate state space in addition to the constraint. Our work forms a similar general dynamic programming framework for the more complex MDP setting.

**Constrained RL.** It is known that stochastic expectation-constrained policies are polynomial-time computable via linear programming [3], and many planning and learning algorithms exist for them [39, 46, 8, 28]. Some learning algorithms can even avoid violation during the learning process under certain assumptions [48, 4]. Furthermore, Brantley et al. [10] developed no-regret algorithms for cMDPs and extended their algorithms to the setting with a constraint on the cost accumulated over all episodes, which is called a knapsack constraint [10, 13].

**Safe RL.** The safe RL community [22, 25] has mainly focused on no-violation learning for stochastic expectation-constrained policies [14, 9, 2, 12, 5] and solving chance constraints [47, 53], which capture the probability of entering unsafe states. Performing learning while avoiding dangerous states [53] is a special case of expectation constraints that has also been studied [42, 44] under

non-trivial assumptions. In addition, instantaneous constraints, which require the expected cost to be within budget at all times, have also been studied [33, 20, 24].

## 2 Cost criteria

In this section, we formalize our problem setting. We also define our conditions for cost criteria.

**Constrained Markov decision processes.** A (tabular, finite-horizon) *Constrained Markov Decision Process* (cMDP) is a tuple $M = (\mathcal{S}, \mathcal{A}, P, r, c, H)$, where (i) $\mathcal{S}$ is the finite set of *states*, (ii) $\mathcal{A}$ is the finite set of *actions*, (iii) $P_h(s, a) \in \Delta(S)$ is the *transition* distribution, (iv) $r_h(s, a) \in \mathbb{R}$ is the *reward*, (v) $c_h(s, a) \in \mathbb{R}$ is the *cost*, and (vi) $H \in \mathbb{N}$ is the finite *time horizon*. We let $S := |\mathcal{S}|$, $A := |\mathcal{A}|$, $[H] := \{1, \ldots, H\}$, and $\mathcal{M}$ denote the set of all cMDPs. We also let $r_{max} \overset{\text{def}}{=} \max_{h,s,a} |r_h(s, a)|$ denote the maximum magnitude reward, $r_{min} \overset{\text{def}}{=} \min_{h,s,a} r_h(s, a)$ denote the true minimum reward, and $p_{min} \overset{\text{def}}{=} \min_{h,s,a,s'} P_h(s' \mid s, a)$ denote the minimum transition probability. Since $\mathcal{S}$ is a finite set, we often assume $\mathcal{S} = [S]$ WLOG. Lastly, for any predicate $p$, we use the Iverson bracket notation $[p]$ to denote 1 if $p$ is true and 0 otherwise, and we let $\chi_p$ denote the characteristic function which evaluates to 0 if $p$ is true and $\infty$ otherwise.

**Interaction protocol.** The agent interacts with $M$ using a policy $\pi = (\pi_h)_{h=1}^H$. In the fullest generality, $\pi_h : \mathcal{H}_h \to \Delta(\mathcal{A})$ is a mapping from the observed history at time $h$ to a distribution of actions. In contrast, a deterministic policy takes the form $\pi_h : \mathcal{H}_h \to \mathcal{A}$. We let $\Pi$ denote the set of all possible policies and $\Pi^D$ denote the set of all deterministic policies. The agent starts at the initial state $s_0 \in \mathcal{S}$ with observed history $\tau_1 = (s_0)$. For any $h \in [H]$, the agent chooses an action $a_h \sim \pi_h(\tau_h)$. Then, the agent receives immediate reward $r_h(s_h, a_h)$ and cost $c_h(s_h, a_h)$. Lastly, $M$ transitions to state $s_{h+1} \sim P_h(s_h, a_h)$ and the agent updates the history to $\tau_{h+1} = (\tau_h, a_h, s_{h+1})$. This process is repeated for $H$ steps; the interaction ends once $s_{H+1}$ is reached.

**Objective.** For any cost criterion $C : \mathcal{M} \times \Pi \to \mathbb{R}$ and budget $B \in \mathbb{R}$, the agent's goal is to compute a solution to the following optimization problem:

$$\max_{\pi \in \Pi} \mathbb{E}_M^\pi \left[ \sum_{h=1}^H r_h(s_h, a_h) \right] \quad \text{s.t.} \quad \begin{cases} C_M^\pi \leq B \\ \pi \text{ deterministic} \end{cases} . \tag{CON}$$

Here, $\mathbb{P}_M^\pi$ denotes the probability law over histories induced from the interaction of $\pi$ with $M$, and $\mathbb{E}_M^\pi$ denotes the expectation defined by this law. We let $V_M^\pi \overset{\text{def}}{=} \mathbb{E}_M^\pi \left[ \sum_{t=1}^H r_t(s_t, a_t) \right]$ denote the value of a policy $\pi$, and $V_M^*$ denote the optimal solution value to (CON).

**Cost criteria.** We consider a broad family of cost criteria that satisfy a strengthening of the standard policy evaluation equations [41]. This strengthening requires not only the cost of a policy to be computable recursively in the time horizon, but at each time the cost should also break down recursively in (state) space.

**Definition 1** (TSR). We call a cost criterion $C$ *time-recursive* (TR) if for any cMDP $M$ and policy $\pi \in \Pi^D$, $\pi$'s cost decomposes recursively into $C_M^\pi = C_1^\pi(s_0)$. Here, $C_{H+1}^\pi(\cdot) = \mathbf{0}$ and for any $h \in [H]$ and $\tau_h \in \mathcal{H}_h$,

$$C_h^\pi(\tau_h) = c_h(s, a) + f\left( \left( P_h(s' \mid s, a), C_{h+1}^\pi(\tau_h, a, s') \right)_{s' \in P_h(s,a)} \right), \tag{TR}$$

where $s = s_h(\tau_h)$, $a = \pi_h(\tau_h)$, and $f$ is a non-decreasing function[1] computable in $O(S)$ time. For technical reasons, we also require that $f(x) = \infty$ whenever $\infty \in x$.

We further say $C$ is *time-space-recursive* (TSR) if the $f$ term above is equal to $g_h^{\tau_h, a}(1)$. Here, $g_h^{\tau_h, a}(S + 1) = 0$ and for any $t \leq S$,

$$g_h^{\tau_h, a}(t) = \alpha \left( \beta \left( P_h(t \mid s, a), C_{h+1}^\pi(\tau_h, a, t) \right), g_h^{\tau_h, a}(t + 1) \right), \tag{SR}$$

---

[1] When we say a multivariate function is non-decreasing, we mean it is non-decreasing with respect to the partial ordering induced by component-wise ordering.

where $\alpha$ is a non-decreasing function, and both $\alpha, \beta$ are computable in $O(1)$ time. We also assume that $\alpha(\cdot, \infty) = \infty$, and $\beta$ satisfies $\alpha(\beta(0, \cdot), x) = x$ to match $f$'s condition.

Since the TR condition is a slight generalization of traditional policy evaluation, it is easy to see that we can solve for minimum-cost policies using backward induction.

**Proposition 1** (TR Intuition). *If $C$ is TR, then $C$ satisfies the usual optimality equations. Furthermore, $\arg\min_{\pi \in \Pi^D} C_M^\pi$ can be computed using backward induction in $O(HS^2A)$ time.*

Although the TR condition is straightforward, the TSR condition is more strict. We will see the utility of the TSR condition in Section 4 when computing Bellman updates. For now, we point out that the TSR condition is not too restrictive: it is satisfied by many popular criteria studied in the literature.

**Proposition 2** (TSR examples). *The following classical constraints can be modeled by a TSR cost constraint.*

1. *(Expectation Constraints) are captured by $C_M^\pi \stackrel{def}{=} \mathbb{E}_M^\pi \left[ \sum_{h=1}^H c_h(s_h, a_h) \right] \leq B$. We see $C$ is TSR by defining $\alpha(x, y) \stackrel{def}{=} x + y$ and $\beta(x, y) \stackrel{def}{=} xy$.*

2. *(Almost Sure Constraints) are captured by $C_M^\pi \stackrel{def}{=} \max_{\substack{\tau \in \mathcal{H}_{H+1}, \\ \mathbb{P}_M^\pi[\tau] > 0}} \sum_{h=1}^H c_h(s_h, a_h) \leq B$. We see $C$ is TSR by defining $\alpha(x, y) \stackrel{def}{=} \max(x, y)$ and $\beta(x, y) \stackrel{def}{=} [x > 0]y$.*

3. *(Anytime Constraints) are captured by $C_M^\pi \stackrel{def}{=} \max_{t \in [H]} \max_{\substack{\tau \in \mathcal{H}_{H+1}, \\ \mathbb{P}_M^\pi[\tau] > 0}} \sum_{h=1}^t c_h(s_h, a_h) \leq B$. We see $C$ is TSR by defining $\alpha(x, y) \stackrel{def}{=} \max(0, \max(x, y))$ and $\beta(x, y) \stackrel{def}{=} [x > 0]y$.*

*Remark* 1 (Extensions). Our methods can also handle stochastic costs and infinite discounting. We defer the details to Appendix F. Moreover, we can handle multiple constraints using vector-valued criteria so long as the comparison operator is a total ordering of the vector space.

*Remark* 2 (Inapproximability). Our methods cannot handle chance constraints or more than one of our example constraints. However, this is not a limitation of our framework as the problem becomes provably inapproximable under said constraints [51, 36].

## 3 Covering algorithm

In this section, we propose an algorithm to solve (CON). Our approach relies on converting the original problem into an equivalent covering problem that can be solved using an unconstrained MDP. This covering MDP is derived using the key idea of value augmentation.

**Packing and covering.** We can view (CON) as a *packing program*, which wishes to maximize $V_M^\pi$ subject to $C_M^\pi \leq B$. However, we could also tackle the problem by reversing the objective: attempt to minimize $C_M^\pi$ subject to $V_M^\pi \geq V_M^*$. If (CON) is feasible, then any optimal solution $\pi$ to this *covering program* satisfies $V_M^\pi \geq V_M^*$ and $C_M^\pi \leq B$. Thus, we can solve the original packing program by solving the covering program.

**Proposition 3** (Packing-Covering Reduction). *Suppose that $C_M^* \stackrel{def}{=} \min_{\pi \in \Pi^D} C_M^\pi$ s.t. $V_M^\pi \geq V_M^*$. Then, $C_M^* \leq B \iff V_M^* > -\infty$. Furthermore, if $V_M^* > -\infty$, then,*

$$\begin{array}{cc} \arg\min_{\pi \in \Pi^D} C_M^\pi \\ V_M^\pi \geq V_M^* \end{array} \subseteq \begin{array}{c} \arg\max_{\pi \in \Pi^D} V_M^\pi \\ C_M^\pi \leq B \end{array}. \tag{PC}$$

*Thus, any solution to the covering program is a solution to the packing program.*

We focus on the covering program for several reasons. To optimize the value recursively, we would need to predict the final cost resulting from intermediate decisions to ensure feasibility. Generally, such predictions would require strict assumptions on the cost criteria. By treating the value as the constraint instead, we only need to assume the cost can be optimized efficiently. Moreover, values are well understood in RL and are more amenable to approximation [36]. Thus, the covering program allows us to capture many criteria, ensure feasibility, and compute accurate value approximations.

---

**Algorithm 1** Reduction to RL

---

**Input:** $(M, C, B)$
1: $\bar{M}, \bar{C} \leftarrow$ Definition 2$(M, C)$
2: $\pi, \bar{C}^* \leftarrow$ SOLVE$(\bar{M}, \bar{C})$
3: **if** $\bar{C}_1^*(s_0, v) > B$ for all $v \in \mathcal{V}$ **then**
4:     **return** "Infeasible"
5: **else**
6:     **return** $\pi$

---

**Value augmentation.** We can solve the covering program by solving a cost-minimizing MDP $\bar{M}$. The key idea is to augment the state space with value demands, $(s, v)$. Then, the agent can recursively reason how to minimize its cost while meeting the current value demand. If the agent starts at $(s_0, V_M^*)$, then an optimal policy for $\bar{M}$ should be a solution to the covering program.

The key invariant we desire is that any feasible policy $\pi$ for $\bar{M}$ should satisfy $\bar{V}_h^\pi(s, v) \geq v$. To ensure this invariance, we recall the policy evaluation equations [41]. If $\pi_h(s) = a$, then,

$$\bar{V}_h^\pi(s, v) = r_h(s, a) + \sum_{s'} P_h(s' \mid s, a) \bar{V}_{h+1}^\pi(s', v_{s'}). \tag{PE}$$

For the value invariant to be satisfied, it suffices for the agent to choose an action $a$ and commit to future value demands $v_{s'}$ satisfying,

$$r_h(s, a) + \sum_{s'} P_h(s' \mid s, a) v_{s'} \geq v. \tag{DEM}$$

We can view choosing future value demands as part of the agent's augmented actions. Then, at any augmented state $(s, v)$, the agent's augmented action space includes all $(a, \mathbf{v}) \in \mathcal{A} \times \mathbb{R}^S$ satisfying (DEM). When $M$ transitions to $s' \sim P_h(s, a)$, the agent's new augmented state should consist of the environment's new state in addition to its chosen demand for that state, $(s', v_{s'})$. Putting these pieces together yields the definition of the cover MDP, Definition 2.

**Definition 2** (Cover MDP). The *cover MDP* $\bar{M} \stackrel{\text{def}}{=} (\bar{\mathcal{S}}, \bar{\mathcal{A}}, \bar{P}, \bar{c}, H)$ where,

1. $\bar{\mathcal{S}} \stackrel{\text{def}}{=} \mathcal{S} \times \mathcal{V}$ where $\mathcal{V} \stackrel{\text{def}}{=} \{v \mid \exists \pi \in \Pi^D, h \in [H+1], \tau_h \in \mathcal{H}_h, V_h^\pi(\tau_h) = v\}$

2. $\bar{\mathcal{A}}_h(s, v) \stackrel{\text{def}}{=} \{(a, \mathbf{v}) \in \mathcal{A} \times \mathcal{V}^S \mid r_h(s, a) + \sum_{s'} P_h(s' \mid s, a) v_{s'} \geq v\}$.

3. $\bar{P}_h((s', v') \mid (s, v), (a, \mathbf{v})) \stackrel{\text{def}}{=} P_h(s' \mid s, a)[v' = v_{s'}]$.

4. $\bar{c}_h((s, v), (a, \mathbf{v})) \stackrel{\text{def}}{=} c_h(s, a)$.

The objective for $\bar{M}$ is to minimize the cost function $\bar{C} \stackrel{\text{def}}{=} C_{\bar{M}}$ with modified base case $\bar{C}_{H+1}^\pi(s, v) \stackrel{\text{def}}{=} \chi_{\{v \leq 0\}}$.

**Covering algorithm.** Importantly, the action space definition ensures the value constraint is satisfied. Meanwhile, the minimum cost objective ensures optimal cost. So long as our cost is TR, $\bar{M}$ can be solved using fast RL methods instead of the brute force computation required for general covering programs. These properties ensure our method, Algorithm 1, is correct.

**Theorem 1** (Reduction). *If* SOLVE *is any finite-time MDP solver, then Algorithm 1 correctly solves* (CON) *in finite time for any TR cost criterion.*

*Remark* 3 (Execution). Given a value-augmented policy $\pi$ output from Algorithm 1, the agent can execute $\pi$ using Algorithm 2. To compute $V_M^*$ as the starting value, it suffices for the agent to compute,

$$V_M^* = \max \{v \in \mathcal{V} \mid \bar{C}_1^*(s_0, v) \leq B\}. \tag{1}$$

This computation can be easily done given $\bar{C}_1^*(s_0, \cdot)$ in $O(|\mathcal{V}|)$ time.

---
**Algorithm 2** Augmented interaction
---
**Input:** $\pi$
 1: $\bar{s}_1 = (s_0, V_M^*)$
 2: **for** $h \leftarrow 1$ to $H$ **do**
 3: $\quad (a, \mathbf{v}) \leftarrow \pi_h(\bar{s}_h)$
 4: $\quad r_h = r_h(s, a)$ and $s_{h+1} \sim P_h(s_h, a)$
 5: $\quad \bar{s}_{h+1} = (s_{h+1}, v_{s_{h+1}})$
---

## 4 Fast Bellman updates

In this section, we present an algorithm to solve $\bar{M}$ from Definition 2 efficiently. Although the Bellman updates can be as hard to solve as the knapsack problem, we use ideas from knapsack approximation algorithms to create an efficient method. Our approach exploits (SR) through approximate dynamic programming on the action space.

Even if $\mathcal{V}$ were small, solving $\bar{M}$ would still be challenging due to the exponentially large action space. Even a single Bellman update requires the solution of a constrained optimization problem:

$$\bar{C}_h^*(s, v) = \min_{a, \mathbf{v}} c_h(s, a) + f\left(\left(P_h(s' \mid s, a), \bar{C}_{h+1}^*\left(s', v_{s'}\right)\right)_{s' \in P_h(s,a)}\right)$$
$$\text{s.t. } r_h(s, a) + \sum_{s'} P_h(s' \mid s, a)v_{s'} \geq v. \tag{BU}$$

Above, we used the fact that $(s', v') \in \bar{P}_h((s, v), (a, \mathbf{v}))$ iff $s' \in P_h(s, a)$ and $v' = v_{s'}$ to simplify $f$'s input. Observe that even when each $v_{s'}$ only takes on two possible values, $\{0, w_{s'}\}$, the optimization above can capture the minimization version of the knapsack problem, implying that it is NP-hard to compute.

**Recursive approach.** Fortunately, we can use the connection to the Knapsack problem positively to efficiently approximate the Bellman update. For any fixed $(s, v) \in \bar{S}$ and $a \in \mathcal{A}$, we focus on the inner constrained minimization over $\mathbf{v}$:

$$\min_{\substack{\mathbf{v} \in \mathcal{V}^S, \\ r_h(s,a) + \sum_{s'} P_h(s' \mid s, a)v_{s'} \geq v}} f\left(\left(P_h(s' \mid s, a), \bar{C}_{h+1}^*\left(s', v_{s'}\right)\right)_{s' \in P_h(s,a)}\right) \tag{2}$$

We use (SR) to transform this minimization over $\mathbf{v}$ into a sequential decision-making problem that decides each $v_{s'}$. As above, we can use the definition of $\bar{P}$ to simplify $g_h^{(s,v),(a,\mathbf{v})}(t, v')$ into a function of $t$ alone:

$$g_h^{(s,v),(a,\mathbf{v})}(t) = \alpha\left(\beta\left(P_h(t \mid s, a), \bar{C}_{h+1}^*(t, v_t)\right), g_h^{(s,v),(a,\mathbf{v})}(t+1)\right). \tag{3}$$

Since $v$ only constrains the valid $(a, \mathbf{v})$ pairs, we can discard $v$ and use the simplified notation $g_{h,\mathbf{v}}^{s,a}(t)$ instead of $g_h^{(s,v),(a,\mathbf{v})}(t)$. It is then clear that we can recursively optimize the value of $v_t$ by focusing on $g_{h,\mathbf{v}}^{s,a}(t)$.

To recursively encode the value constraint, we can record the partial value $u = r_h(s, a) + \sum_{s'=1}^{t-1} P_h(s' \mid s, a)v_{s'}$ that we have accumulated so far. Then, we can check if our choices for $\mathbf{v}$ satisfied the constraint with the inequality $u \geq v$. The formal recursion is defined in Definition 3.

**Definition 3.** For any $h \in [H]$, $s \in \mathcal{S}$, $v \in \mathcal{V}$, and $u \in \mathbb{R}$, we define, $g_{h,v}^{s,a}(S + 1, u) = \chi_{\{u \geq v\}}$ and for $t \leq S$,

$$g_{h,v}^{s,a}(t, u) = \min_{v_t \in \mathcal{V}} \alpha\left(\beta\left(P_h(t \mid s, a), \bar{C}_{h+1}^*(t, v_t)\right), g_{h,v}^{s,a}(t+1, u + P_h(t \mid s, a)v_t)\right). \tag{DP}$$

**Recursive rounding.** This approach can still be slow due to the exponential number of partial values $u$ induced. Similarly to the knapsack problem, the key is to round each input $u$ to ensure fewer subproblems. Unlike the knapsack problem, however, we do not have an easily computable lower bound on the optimal value. Thus, we turn to a more aggressive recursive rounding. Since rounding may cause originally feasible values to violate the demand constraint, we also relax the demand constraint to $u \geq \kappa(v)$ for some lower bound function $\kappa$.

---

**Algorithm 3** Approx Bellman Update

**Input:** $(h, s, v, \bar{C}^*_{h+1})$
1: **for** $a \in \mathcal{A}$ **do**
2:     $\hat{g}^{s,a}_{h,v}(S+1, u) \leftarrow \chi_{\{u \geq v\}} \ \forall u \in \hat{\mathcal{U}}^{s,a}_h(S+1)$
3:     **for** $t \leftarrow S$ down to 1 **do**
4:         **for** $u \in \hat{\mathcal{U}}^{s,a}_h(t)$ **do**
5:             $v_{t,a}, \hat{g}^{s,a}_{h,v}(t, u) \leftarrow$ (ADP)
6:   $a^*, \hat{C}^*_h(s, v) \leftarrow \min_{a \in \mathcal{A}} c_h(s, a) + \hat{g}^{s,a}_{h,v}(1, r_h(s, a))$
7: **return** $(a^*, v_{1,a^*}, \ldots, v_{S,a^*})$ and $\hat{C}^*_h(s, v)$

---

**Algorithm 4** Approx Solve

**Input:** $(\bar{M}, \bar{C})$
1: $\hat{C}^*_{H+1}(s, v) \leftarrow \chi_{\{v \leq 0\}}$ for all $(s, v) \in \bar{\mathcal{S}}$
2: **for** $h \leftarrow H$ down to 1 **do**
3:     **for** $(s, v) \in \bar{\mathcal{S}}$ **do**
4:         $\hat{a}, \hat{C}^*_h(s, v) \leftarrow$ Algorithm 3$(h, s, v, \hat{C}^*_{h+1})$
5:         $\pi_h(s, v) \leftarrow \hat{a}$
6: **return** $\pi$ and $\hat{C}^*$

---

**Definition 4.** Fix a rounding function $\lfloor \cdot \rfloor_{\mathcal{G}}$ and a lower bound function $\kappa$. For any $h \in [H]$, $s \in \mathcal{S}$, $v \in \mathcal{V}$, and $u \in \mathbb{R}$, we define, $\hat{g}^{s,a}_{h,v}(S+1, u) = \chi_{\{u \geq v\}}$ and for $t \leq S$,

$$\hat{g}^{s,a}_{h,v}(t, u) \stackrel{def}{=} \min_{v_t \in \mathcal{V}} \alpha \left( \beta \left( P_h(t \mid s, a), \bar{C}^*_{h+1}(t, v_t) \right), \hat{g}^{s,a}_{h,v}(t+1, \lfloor u + P_h(t \mid s, a)v_t \rfloor_{\mathcal{G}}) \right). \quad \text{(ADP)}$$

Fortunately, the approximate version behaves similarly to the original. The main difference is the constraint now ensures the rounded sums are at least the value lower bound. This is formalized in Lemma 1.

**Lemma 1.** *For any $t \in [S+1]$ and $u \in \mathbb{R}$, we have that,*

$$\hat{g}^{s,a}_{h,v}(t, u) = \min_{\mathbf{v} \in \mathcal{V}^{S-t+1}} g^{s,a}_{h,\hat{\mathbf{v}}}(t) \tag{4}$$
$$\text{s.t.} \quad \hat{\sigma}^{s,a}_{h,\mathbf{v}}(t, u) \geq \kappa(v),$$

*where $\hat{\sigma}^{s,a}_{h,\mathbf{v}}(t, u) \stackrel{def}{=} \lfloor \lfloor u + P_h(t \mid s, a)v_t \rfloor_{\mathcal{G}} + \ldots + P_h(S \mid s, a)v_S \rfloor_{\mathcal{G}}.$*

To turn this recursion into a usable dynamic programming algorithm, we must also pre-compute the inputs to any sub-computation. Unlike in standard RL, this computation must be done with a forward recursion. The details for the approximate Bellman update are given in Definition 5.

**Definition 5** (Approx Bellman). For any $h \in [H]$, $s \in \mathcal{S}$, and $a \in \mathcal{A}$, we define $\hat{\mathcal{U}}^{s,a}_h(1) \stackrel{def}{=} \{r_h(s, a)\}$ and for any $t \in [S]$,

$$\hat{\mathcal{U}}^{s,a}_h(t+1) \stackrel{def}{=} \bigcup_{v_t \in \mathcal{V}} \bigcup_{u \in \hat{\mathcal{U}}^{s,a}_h(t)} \left\{ \lfloor u + P_h(t \mid s, a)v_t \rfloor_{\mathcal{G}} \right\}. \tag{5}$$

Then, an approximation to the Bellman update can be computed using Algorithm 3.[2]

**Proposition 4.** *Algorithm 4 runs in $O(HS^2 A|\mathcal{V}|^2 \hat{U})$ time, where $\hat{U} \stackrel{def}{=} \max_{h,s,a} |\hat{\mathcal{U}}^{s,a}_h|$. When $\lfloor \cdot \rfloor_{\mathcal{G}}$ and $\kappa$ are the identity function, Algorithm 4 outputs an optimal solution to $\bar{M}$.*

*Remark* 4 (Speedups). The runtime of our methods can be quadratically improved by rounding the differences instead of the sums. We defer the details to Appendix F.

---

[2]We use the notation $x, o \leftarrow \min_x z(x)$ to say that $x$ is the minimizer and $o$ the value of the optimization.

# 5 Approximation algorithms

In this section, we present our approximation algorithms for solving (CON). We carefully round the value demands over both time and space to induce an approximate MDP. Solving this approximate MDP with Algorithm 4 yields our FPTAS.

Although we can avoid exponential-time Bellman updates, the running time of the approximate Bellman update will still be slow if $|\mathcal{V}|$ is large. To reduce the complexity, we instead use a smaller set of approximate values by rounding elements of $|\mathcal{V}|$. By rounding down, we effectively relax the value-demand constraint. More aggressive rounding not only leads to smaller augmented state spaces but also to smaller cost policies. The trade-off is aggressive rounding leads to weaker guarantees on the computed policy's value. Thus, it is critical to carefully design the rounding and lower bound functions to balance this trade-off.

**Value approximation.** Given a rounding down function $\lfloor \cdot \rfloor_{\mathcal{G}}$, we would ideally use the rounded set $\left\{ \lfloor v \rfloor_{\mathcal{G}} \mid v \in \mathcal{V} \right\}$ to form our approximate state space. To avoid having to compute $\mathcal{V}$ explicitly, we instead use the rounded superset $\left\{ \lfloor v \rfloor_{\mathcal{G}} \mid v \in [v_{min}, v_{max}] \right\}$, where $v_{min}$ and $v_{max}$ are bounds on the extremal values that we specify later. To ensure we can use Algorithm 4 to find solutions efficiently, we must also relax the augmented action space to only include vectors that lead to feasible subproblems for (ADP). From Lemma 1, we know this is exactly the set of $(a, \hat{\mathbf{v}})$ for which $\hat{\sigma}_{h,\hat{\mathbf{v}}}^{s,a}(1, r_h(s,a)) \geq \kappa(v)$. Combining these ideas yields the new approximate MDP, defined in Definition 6.

**Definition 6** (Approximate MDP). Given a rounding function $\lfloor \cdot \rfloor_{\mathcal{G}}$ and lower bound function $\kappa$, the *approximate MDP* $\hat{M} \stackrel{\text{def}}{=} (\hat{\mathcal{S}}, \hat{\mathcal{A}}, \hat{P}, \hat{c}, H)$ where,

1. $\hat{\mathcal{S}} \stackrel{\text{def}}{=} \mathcal{S} \times \hat{\mathcal{V}}$ where $\hat{\mathcal{V}} \stackrel{\text{def}}{=} \left\{ \lfloor v \rfloor_{\mathcal{G}} \mid v \in [v_{min}, v_{max}] \right\}$.

2. $\hat{\mathcal{A}}_h(s, \hat{v}) \stackrel{\text{def}}{=} \left\{ (a, \hat{\mathbf{v}}) \in \mathcal{A} \times \hat{\mathcal{V}}^S \mid \hat{\sigma}_{h,\hat{\mathbf{v}}}^{s,a}(1, r_h(s,a)) \geq \kappa(\hat{v}) \right\}$.

3. $\hat{P}_h((s', \hat{v}') \mid (s, \hat{v}), (a, \hat{\mathbf{v}})) \stackrel{\text{def}}{=} P_h(s' \mid s, a)[\hat{v}' = \hat{v}_{s'}]$.

4. $\hat{c}_h((s, \hat{v}), (a, \hat{\mathbf{v}})) \stackrel{\text{def}}{=} c_h(s, a)$.

The objective for $\hat{M}$ is to minimize the cost function $\hat{C} \stackrel{\text{def}}{=} C_{\hat{M}}$ with modified base case $\hat{C}_{H+1}^{\pi}(s, \hat{v}) \stackrel{\text{def}}{=} \chi_{\{\hat{v} \leq 0\}}$.

We can show that rounding down in Definition 6 achieves our goal of producing smaller cost policies. This ensures feasibility is even easier to achieve. We formalize this observation in Lemma 2.

**Lemma 2** (Optimistic Costs). *For our later choices of $\lfloor \cdot \rfloor_{\mathcal{G}}$ and $\kappa$, the following holds: for any $h \in [H+1]$ and $(s, v) \in \bar{\mathcal{S}}$, we have $\hat{C}_h^*(s, \lfloor v \rfloor_{\mathcal{G}}) \leq \bar{C}_h^*(s, v)$.*

Thus, Algorithm 5 always outputs a policy with better than optimal cost when the instance is feasible, $V_M^* > -\infty$. If the instance is infeasible, all policies have cost larger than $B$ by definition and so Algorithm 5 correctly indicates the instance is infeasible. The remaining question is whether Algorithm 5 outputs policies having near-optimal value.

**Time-Space errors.** To assess the optimality gap of Algorithm 5 policies, we must first explore the error accumulated by our rounding approach. Rounding each value naturally accumulates approximation error over time. Rounding the partial values while running Algorithm 3 accumulates additional error over (state) space. Thus, solving $\hat{M}$ using Algorithm 4 accumulates error over both time and space, unlike other approximate methods in RL. As a result, our rounding and threshold functions will generally depend on both $H$ and $S$.

**Arithmetic rounding.** Our first approach is to round each value down to its closest element in a $\delta$-cover. This guarantees that $v - \delta \leq \lfloor v \rfloor_{\mathcal{G}} \leq v$. Thus, $\lfloor v \rfloor_{\mathcal{G}}$ is an underestimate that is not too far from the true value. By setting $\delta$ to be inversely proportional to $SH$, we control the errors over time and space. The lower bound must also be a function of $S$ since it controls the error over space.

---

**Algorithm 5** Approximation Scheme

---

**Input:** $(M, C, B)$

1: **Hyperparameters:** $\lfloor \cdot \rfloor_{\mathcal{G}}$ and $\kappa$
2: $\hat{M}, \hat{C} \leftarrow$ Definition 6$(M, C, \lfloor \cdot \rfloor_{\mathcal{G}}, \kappa)$
3: $\pi, \hat{C}^* \leftarrow$ Algorithm 4$(\hat{M}, \hat{C})$
4: **if** $\hat{C}_1^*(s_0, \hat{v}) > B$ for all $\hat{v} \in \hat{\mathcal{V}}$ **then**
5:     **return** "Infeasible"
6: **else**
7:     **return** $\pi$

---

**Definition 7** (Additive Approx). Fix $\epsilon > 0$. We define,

$$\lfloor v \rfloor_{\mathcal{G}} \overset{\text{def}}{=} \left\lfloor \frac{v}{\delta} \right\rfloor \delta \text{ and } \kappa(v) \overset{\text{def}}{=} v - \delta(S+1), \tag{6}$$

where $\delta \overset{\text{def}}{=} \frac{\epsilon}{H(S+1)+1}$, $v_{min} \overset{\text{def}}{=} -Hr_{max}$, and $v_{max} \overset{\text{def}}{=} Hr_{max}$.

**Theorem 2** (Additive FPTAS). *For any $\epsilon > 0$, Algorithm 5 using Definition 7 given any cMDP $M$ and TSR criteria $C$ either correctly outputs the instance is infeasible, or produces a policy $\pi$ satisfying $\hat{V}^{\pi} \geq V_M^* - \epsilon$ in $O(H^7 S^5 A r_{max}^3 / \epsilon^3)$ time. Thus, it is an additive-FPTAS for the class of cMDPs with polynomial-bounded $r_{max}$ and TSR criteria.*

**Geometric rounding.** Since the arithmetic approach can be slow when $r_{max}$ is large, we can instead round values down to their closest power of $1/(1 - \delta)$. This guarantees the number of approximate values needed is upper bounded by a function of $\log(r_{max})$, which is polynomial in the input size. We choose a geometric scheme satisfying $v(1 - \delta) \leq \lfloor v \rfloor_{\mathcal{G}} \leq v$ so that the rounded value is an underestimate and a relative approximation to the true value. To ensure this property, we must now require that all rewards are non-negative.

**Definition 8** (Relative Approx). Fix $\epsilon > 0$. We define,

$$\lfloor v \rfloor_{\mathcal{G}} \overset{\text{def}}{=} v^{min} \left( \frac{1}{1-\delta} \right)^{\left\lfloor \log_{\frac{1}{1-\delta}} \frac{v}{v^{min}} \right\rfloor} \text{ and } \kappa(v) \overset{\text{def}}{=} v(1-\delta)^{S+1}, \tag{7}$$

where $\delta \overset{\text{def}}{=} \frac{\epsilon}{H(S+1)+1}$, $v_{min} = p_{min}^H r_{min}$, and $v_{max} = Hr_{max}$.

**Theorem 3** (Relative FPTAS). *For $\epsilon > 0$, Algorithm 5 using Definition 8 given any cMDP $M$ and TSR criteria $C$ either correctly outputs the instance is infeasible, or produces a policy $\pi$ satisfying $\hat{V}^{\pi} \geq V_M^*(1 - \epsilon)$ in $O(H^7 S^5 A \log(r_{max}/r_{min}p_{min})^3 / \epsilon^3)$ time. Thus, it is a relative-FPTAS for the class of cMDPs with non-negative rewards and TSR criteria.*

*Remark* 5 (Assumption Necessity). We also note the mild reward assumptions we made to guarantee efficiency are unavoidable. Without reward bounds, (CON) captures the knapsack problem which does not admit additive approximations. Similarly, without non-negativity, relative approximations for maximization problems are generally not computable.

## 6 Conclusions

In this paper, we studied the computational complexity of computing deterministic policies for CRL problems. Our main contribution was the design of an FPTAS, Algorithm 5, that solves (CON) for any cMPD and TSR criteria under mild reward assumptions. In particular, our method is an additive-FPTAS if the cMDP's rewards are polynomially bounded, and is a relative-FPTAS if the cMDP's rewards are non-negative. We note these assumptions are necessary for efficient approximation, so our algorithm achieves the best approximation guarantees possible under worst-case analysis. Moreover, our algorithmic approach, which uses approximate dynamic programming over time and the state space, highlights the importance of the TSR condition in making (CON) tractable. Our work finally resolves the long-standing open questions of polynomial-time approximability for 1) anytime-constrained policies, 2) almost-sure-constrained policies, and 3) deterministic expectation-constrained policies.

**Future work.** Several interesting questions remain unanswered. First, it remains unresolved whether an FPTAS exists asymptotically faster than ours. Second, whether our TSR condition is necessary for efficient computation or whether a more general condition could be derived is unclear. Lastly, it is open whether there exist algorithms that can feasibly handle multiple constraints from Proposition 2. Although computing feasible policies for multiple constraints is NP-hard, special cases may be approximable efficiently. Moreover, an average-case or smoothed-case analysis could circumvent this worst-case hardness.

## Acknowledgments and Disclosure of Funding

This work was supported in part by NSF grant 2023239.

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

## A    Appendix / supplemental material

# B Proofs for Section 2

## B.1 Proof of Proposition 1

The proof follows from the standard proof of backward induction [41]. The main ideas for the proof can also be seen in the proof of Lemma 4 and Lemma 5.

## B.2 Proof of Proposition 2

*Proof.*

1. (Expectation Constraints) We claim that $C_M^\pi$ captures expectation constraints. This is immediate as an expectation constraint takes the form $\mathbb{E}_M^\pi \left[ \sum_{h=1}^H c_h(s_h, a_h) \right] \leq B$ and by definition $C_M^\pi = \mathbb{E}_M^\pi \left[ \sum_{h=1}^H c_h(s_h, a_h) \right]$. Moreover, the standard policy evaluation equations for deterministic policies immediately imply,

   $$C_h^\pi(\tau_h) = c_h(s, a) + \sum_{s'} P_h(s' \mid s, a) C_{h+1}^\pi(\tau_h, a, s'). \tag{EC}$$

   Thus, (TR) holds. It is also easy to see that $\sum_{s'} P_h(s' \mid s, a) C_{h+1}^\pi(\tau_h, a, s')$ can be computed recursively state-wise by,

   $$P_h(1 \mid s, a) C_{h+1}^\pi(\tau_h, a, 1) + \sum_{s'=2}^S P_h(s' \mid s, a) C_{h+1}^\pi(\tau_h, a, s'), \tag{8}$$

   and so (SR) holds. The infinity conditions and non-decreasing requirements are also easy to verify.

2. (Almost Sure Constraints) We claim that $C_M^\pi$ captures almost sure constraints. This is because that for tabular MDPs, $\mathbb{P}_M^\pi[\sum_{h=1}^H c_h(s_h, a_h) \leq B] = 1$ if and only if for all $\tau \in \mathcal{H}_{H+1}$ with $\mathbb{P}_M^\pi[\tau] > 0$ it holds that $\sum_{h=1}^H c_h(s_h, a_h) \leq B$ if and only if $C_M^\pi = \max_{\substack{\tau \in \mathcal{H}_{H+1}: \\ \mathbb{P}_M^\pi[\tau] > 0}} \sum_{h=1}^H c_h(s_h, a_h) \leq B$.

   Let $c(\tau) = \sum_{h=1}^H c_h(s_h, a_h)$ denote the cost of a full history $\tau \in \mathcal{H}_{H+1}$ and let $c_{h:t}(\tau) = \sum_{k=h}^t c_k(s_k, a_k)$ denote the partial cost of $\tau$ from time $h$ to time $t$. Our choice of $\alpha$ and $\beta$ imply that,

   $$C_h^\pi(\tau_h) = c_h(s, a) + \max_{s' \in P_h(s,a)} C_{h+1}^\pi(\tau_h, a, s'). \tag{ASC}$$

   To show that $C_M^\pi$ satisfies (TR), we prove for all $h \in [H+1]$ and all $\tau_h \in \mathcal{H}_h$ that

   $$C_h(\tau_h) = \max_{\substack{\tau \in \mathcal{H}_{H+1}: \\ \mathbb{P}_M^\pi[\tau \mid \tau_h] > 0}} c_{h:H}(\tau). \tag{9}$$

   Then, we see that $C_1^\pi(s_0) = \max_{\substack{\tau \in \mathcal{H}_{H+1}: \\ \mathbb{P}_M^\pi[\tau \mid s_0] > 0}} c_{1:H}(\tau) = \max_{\substack{\tau \in \mathcal{H}_{H+1}: \\ \mathbb{P}_M^\pi[\tau] > 0}} \sum_{h=1}^H c_h(s_h, a_h) = C_M^\pi$. Thus, $C_M^\pi$ satisfies (TR). Furthermore, it is clear that $\max_{s' \in P_h(s,a)} C_{h+1}^\pi(\tau_h, a, s')$ can be computed state-recursively by,

   $$\max(C_{h+1}^\pi(\tau_h, a, 1)[P_h(1 \mid s, a) > 0], \max_{s'=2}^S C_{h+1}^\pi(\tau_h, a, s')[P_h(s' \mid s, a) > 0]), \tag{10}$$

   and so $C_M^\pi$ satisfies (SR). The infinity conditions and non-decreasing requirements are also easy to verify.

   We proceed by induction on $h$.

   - (Base Case) For the base case, we consider $h = H + 1$. Observe that for any history $\tau$, we have $c_{H+1:H}(\tau) = 0$ since it is an empty sum. Then, by definition of $C_M^\pi$, we see that $C_{H+1}^\pi(\tau_{H+1}) = 0 = \max_\tau 0 = \max_\tau c_{H+1:H}(\tau)$.

- (Inductive Step) For the inductive step, we consider $h \leq H$. Let $s = s_h(\tau_h)$ and $a = \pi_h(\tau_h)$. For any $\tau \in \mathcal{H}_{H+1}$ for which $\mathbb{P}_M^\pi[\tau \mid \tau_h] > 0$, we can decompose its cost by $c_{h:H}(\tau) = c_h(s, a) + c_{h+1:H}(\tau)$. Since $a$ is fixed, we can remove $c_h(s, a)$ from the optimization to get,

$$\max_{\substack{\tau \in \mathcal{H}_{H+1}: \\ \mathbb{P}_M^\pi[\tau|\tau_h]>0}} c_{h:H}(\tau) = c_h(s, a) + \max_{\substack{\tau \in \mathcal{H}_{H+1}: \\ \mathbb{P}_M^\pi[\tau|\tau_h]>0}} c_{h+1:H}(\tau).$$

Next, we observe by the Markov property that $\mathbb{P}_M^\pi[\tau \mid \tau_h] = \sum_{s'} \mathbb{P}_M^\pi[\tau \mid \tau_h, a, s']P_h(s' \mid s, a)$. Thus, $\mathbb{P}_M^\pi[\tau \mid \tau_h] > 0$ if and only if there exists some $s' \in P_h(s, a)$ satisfying $\mathbb{P}_M^\pi[\tau \mid \tau_h, a, s'] > 0$. This implies that,

$$\max_{\substack{\tau \in \mathcal{H}_{H+1}: \\ \mathbb{P}_M^\pi[\tau|\tau_h]>0}} c_{h+1:H}(\tau) = \max_{s' \in P_h(s,a)} \max_{\substack{\tau \in \mathcal{H}_{H+1}: \\ \mathbb{P}_M^\pi[\tau|\tau_h,a,s']>0}} c_{h+1:H}(\tau).$$

By applying the induction hypothesis, we see that,

$$\max_{\substack{\tau \in \mathcal{H}_{H+1}: \\ \mathbb{P}_M^\pi[\tau|\tau_h]>0}} c_{h:H}(\tau) = c_h(s, a) + \max_{s' \in P_h(s,a)} \max_{\substack{\tau \in \mathcal{H}_{H+1}: \\ \mathbb{P}_M^\pi[\tau|\tau_h,a,s']>0}} c_{h+1:H}(\tau)$$

$$= c_h(s, a) + \max_{s' \in P_h(s,a)} C_{h+1}^\pi(\tau_h, a, s')$$

$$= C_h(\tau_h).$$

The second line used the induction hypothesis and the third line used the definition of $C_M^\pi$.

3. (Anytime Constraints) We claim that $C_M^\pi$ captures anytime constraints. This is because that for tabular MDPs, $\mathbb{P}_M^\pi[\forall t \in [H], \sum_{h=1}^t c_h(s_h, a_h) \leq B] = 1$ if and only if for all $t \in [H]$ and $\tau \in \mathcal{H}_{H+1}$ with $\mathbb{P}_M^\pi[\tau] > 0$ it holds that $\sum_{h=1}^t c_h(s_h, a_h) \leq B$ if and only if $C_M^\pi = \max_{t \in [H]} \max_{\tau \in \mathcal{H}_{H+1}:\mathbb{P}_M^\pi[\tau]>0} \sum_{h=1}^t c_h(s_h, a_h) \leq B$.

Our choice of $\alpha$ and $\beta$ imply that,

$$C_h^\pi(\tau_h) = c_h(s, a) + \max\left(0, \max_{s' \in P_h(s,a)} C_{h+1}^\pi(\tau_h, a, s')\right). \tag{AC}$$

To show that $C_M^\pi$ satisfies (TR), we show that for all $h \in [H+1]$ and all $\tau_h \in \mathcal{H}_h$ that

$$C_h(\tau_h) = \max_{t \geq h} \max_{\substack{\tau \in \mathcal{H}_{H+1}: \\ \mathbb{P}_M^\pi[\tau|\tau_h]>0}} c_{h:t}(\tau). \tag{11}$$

Then, we see that $C_1^\pi(s_0) = \max_{t \in [H]} \max_{\substack{\tau \in \mathcal{H}_{H+1}: \\ \mathbb{P}_M^\pi[\tau|s_0]>0}} c_{1:t}(\tau) = \max_{t \in [H]} \max_{\substack{\tau \in \mathcal{H}_{H+1}: \\ \mathbb{P}_M^\pi[\tau]>0}}$ $\sum_{h=1}^t c_h(s_h, a_h) = C_M^\pi$. Thus, $C_M^\pi$ satisfies (TR). Furthermore, it is clear that $\max(0, \max_{s' \in P_h(s,a)} C_{h+1}^\pi(\tau_h, a, s'))$ can be computed state-recursively by,

$$\max\Big( \max(0, C_{h+1}^\pi(\tau_h, a, 1)[P_h(1 \mid s, a) > 0]),$$

$$\max(0, \max_{s'=2}^S C_{h+1}^\pi(\tau_h, a, s')[P_h(s' \mid s, a) > 0])\Big), \tag{12}$$

and so $C_M^\pi$ satisfies (SR). The infinity conditions and non-decreasing requirements are also easy to verify.

We proceed by induction on $h$.

- (Base Case) For the base case, we consider $h = H + 1$. Observe that for any history $\tau$ and $t$, we have $c_{H+1:t}(\tau) = 0$ since it is an empty sum. Then, by definition of $C_M^\pi$, we see that $C_{H+1}^\pi(\tau_{H+1}) = 0 = \max_t \max_\tau 0 = \max_t \max_\tau c_{H+1:t}(\tau)$[3].

---

[3] Technically, there is no $t \in [H]$ satisfying $t \geq H + 1$. We instead interpret the $t \geq h$ condition in the max as over all integers and define the immediate costs to be 0 for all future times to simplify the base case.

- (Inductive Step) For the inductive step, we consider $h \leq H$. Let $s = s_h(\tau_h)$ and $a = \pi_h(\tau_h)$. By separately considering the case where $t = h$ and $t \geq h+1$ in the $\max_{t \geq h}$, we see that $\max_{t \geq h} \max_{\substack{\tau \in \mathcal{H}_{H+1}: \\ \mathbb{P}_M^\pi[\tau|\tau_h]>0}} c_{h:t}(\tau)$

$$= \max\left( \max_{\substack{\tau \in \mathcal{H}_{H+1}: \\ \mathbb{P}_M^\pi[\tau|\tau_h]>0}} c_{h:h}(\tau), \max_{t \geq h+1} \max_{\substack{\tau \in \mathcal{H}_{H+1}: \\ \mathbb{P}_M^\pi[\tau|\tau_h]>0}} c_{h:t}(\tau) \right)$$

$$= \max\left( c_h(s,a), c_h(s,a) + \max_{t \geq h+1} \max_{\substack{\tau \in \mathcal{H}_{H+1}: \\ \mathbb{P}_M^\pi[\tau|\tau_h]>0}} c_{h+1:t}(\tau) \right)$$

$$= c_h(s,a) + \max\left( 0, \max_{t \geq h+1} \max_{\substack{\tau \in \mathcal{H}_{H+1}: \\ \mathbb{P}_M^\pi[\tau|\tau_h]>0}} c_{h+1:t}(\tau) \right)$$

$$= c_h(s,a) + \max\left( 0, \max_{t \geq h+1} \max_{s' \in P_h(s,a)} \max_{\substack{\tau \in \mathcal{H}_{H+1}: \\ \mathbb{P}_M^\pi[\tau|\tau_h,a,s']>0}} c_{h+1:t}(\tau) \right)$$

$$= c_h(s,a) + \max\left( 0, \max_{s' \in P_h(s,a)} \max_{t \geq h+1} \max_{\substack{\tau \in \mathcal{H}_{H+1}: \\ \mathbb{P}_M^\pi[\tau|\tau_h,a,s']>0}} c_{h+1:t}(\tau) \right)$$

$$= c_h(s,a) + \max\left( 0, \max_{s' \in P_h(s,a)} C_{h+1}^\pi(\tau_h, a, s') \right)$$

$$= C_h(\tau_h).$$

The second line used the fact that $c_{h:h}(\tau) = c_h(s,a)$ and the recursive definition of $c_{h:t}(\tau)$. The fourth line used the result proven for the almost sure case above. The sixth line used the induction hypothesis. The last line used the definition of $C_M^\pi$.

$\square$

## C   Proofs for Section 3

### C.1   Helpful Technical Lemmas

Here, we use a different, inductive definition for $\mathcal{V}$ then in the main text. However, the following lemma shows they are equivalent.

**Definition 9** (Value Space). *For any $s \in \mathcal{S}$, we define $\mathcal{V}_{H+1}(s) \overset{\text{def}}{=} \{0\}$, and for any $h \in [H]$,*

$$\mathcal{V}_h(s) \overset{\text{def}}{=} \bigcup_a \bigcup_{\mathbf{v} \in \times_{s'} \mathcal{V}_{h+1}(s')} \left\{ r_h(s,a) + \sum_{s'} P_h(s' \mid s, a) v_{s'} \right\}. \tag{13}$$

We define $\mathcal{V} \overset{\text{def}}{=} \bigcup_{h,s} \mathcal{V}_h(s)$.

**Lemma 3** (Value Intuition). *For all $s \in \mathcal{S}$ and $h \in [H+1]$,*

$$\mathcal{V}_h(s) = \left\{ v \in \mathbb{R} \mid \exists \pi \in \Pi^D, \tau_h \in \mathcal{H}_h, \; (s = s_h(\tau_h) \wedge V_h^\pi(\tau_h) = v) \right\}, \tag{14}$$

*and $|\mathcal{V}_h(s)| \leq A^{\sum_{t=h}^H S^{H-t}}$. Thus, $\mathcal{V}$ can be computed in finite time using backward induction.*

**Lemma 4** (Cost). *For any $h \in [H+1]$, $\tau_h \in \mathcal{H}_h$, and $v \in \mathcal{V}$, if $s = s_h(\tau_h)$, then,*

$$\bar{C}_h^*(s, v) \leq \min_{\pi \in \Pi^D} C_h^\pi(\tau_h)$$
$$\text{s.t. } V_h^\pi(\tau_h) \geq v. \tag{15}$$

**Lemma 5** (Value). *Suppose that $\pi \in \Pi^D$. For all $h \in [H+1]$ and $(s,v) \in \bar{\mathcal{S}}$, if $\bar{C}_h^\pi(s,v) < \infty$, then $\bar{V}_h^\pi(s,v) \geq v$.*

*Remark* 6 (Technical Subtlety). Technically, $V_h^\pi(\tau_h)$ is only well defined if $\mathbb{P}_M^\pi[\tau_h] > 0$ and all of our arguments technically should assume this is the case. However, it is standard in MDP theory to define the policy evaluation equations on non-reachable trajectories using the standard recursion to simplify proofs, as we have done here. Formally, this is equivalent to assuming the process starts initially at $\tau_h$ instead of just conditioning on reaching $\tau_h$, or defining the values to correspond to policy evaluation equations directly. This is consistent with the usual definition when $\mathbb{P}_M^\pi[\tau_h] > 0$ but gives it a defined value also when $\mathbb{P}_M^\pi[\tau_h] = 0$. In either case, this detail only means our recursive definition of $\mathcal{V}$ is a superset rather than exactly the set of all values as we defined in the main text. This does not effect the final results since unreachable trajectories do not effect $\pi$'s overall value in the MDP anyway, and only effects the interpretations of some intermediate variables.

## C.2 Proof of Proposition 3

*Proof.* By definition of $V_M^*$ and $C_M^*$,

$$V_M^* > -\infty \iff \exists \pi \in \Pi^D, \ C_M^\pi \leq B \wedge V_M^\pi \geq V_M^*$$
$$\iff C_M^* \leq B.$$

For the second claim, we observe that if $V_M^* > -\infty$ then by the above argument any optimal deterministic policy $\pi$ for COVER satisfies $C_M^\pi = C_M^* \leq B$ and $V_M^\pi \geq V_M^*$. Thus, $COVER \subseteq PACK$. $\qquad\square$

## C.3 Proof of Lemma 3

*Proof.* We proceed by induction on $h$. Let $s \in \mathcal{S}$ be arbitrary.

**Base Case.** For the base case, we consider $h = H+1$. In this case, we know that for any $\pi \in \Pi^D$ and any $\tau \in \mathcal{H}_{H+1}$, $V_{H+1}^\pi(\tau_{H+1}) = 0 \in \{0\} = \mathcal{V}_{H+1}(s)$ by definition. Furthermore, $|\mathcal{V}_{H+1}(s)| = 1 = A^0 = A^{\sum_{t=H+1}^H S^t}$.

**Inductive Step.** For the inductive step, we consider $h \leq H$. In this case, we know that for any $\pi \in \Pi^D$ and any $\tau_h \in \mathcal{H}_h$, if $s = s_h(\tau_h)$ and $a = \pi_h(\tau_h)$, then the policy evaluation equations imply,

$$V_h^\pi(\tau_h) = r_h(s,a) + \sum_{s'} P_h(s' \mid s,a) V_{h+1}^\pi(\tau_h, a, s').$$

We know by the induction hypothesis that $V_{h+1}^\pi(\tau_h, a, s') \in \mathcal{V}_{h+1}(s')$. Thus, by (13), $V_h^\pi(\tau_h) \in \mathcal{V}_h(s)$. Lastly, we see by (13) and the induction hypothesis that,

$$|\mathcal{V}_h(s)| \leq A \prod_{s'} |\mathcal{V}_{h+1}(s')| \leq A \prod_{s'} A^{\sum_{t=h+1}^H S^{H-t}} = A^{1+S\sum_{t=h+1}^H S^{H-t}} = A^{\sum_{t=h}^H S^{H-t}}.$$

This completes the proof. $\qquad\square$

## C.4 Proof of Lemma 4

*Proof.* We proceed by induction on $h$. Let $\tau_h \in \mathcal{H}_h$ and $v \in \mathcal{V}$ be arbitrary and suppose that $s = s_h(\tau_h)$. We let $C_h^*(\tau_h, v)$ denote the minimum for the RHS of (15).

**Base Case.** For the base case, we consider $h = H+1$. Observe that for any $\pi \in \Pi^D$, $V_{H+1}^\pi(\tau_{H+1}) = 0$ by definition. Thus, there exists a $\pi \in \Pi^D$ satisfying $V_{H+1}^\pi(\tau_{H+1}) \geq v$ if and only if $v \leq 0$. We also know by definition that any such policy $\pi$ satisfies $C_{H+1}^\pi(\tau_{H+1}) = 0$ and if no such policy exists $C_{H+1}^*(\tau_{H+1}, v) = \infty$ by convention. Therefore, we see that $C_{H+1}^*(\tau_{H+1}, v) = \chi_{\{v \leq 0\}}$. Then, by definition of the base case for $\bar{C}$, it follows that,

$$\bar{C}_{H+1}^*(s,v) = \chi_{\{v \leq 0\}} = C_{H+1}^*(\tau_{H+1}, v).$$

**Inductive Step.** For the inductive step, we consider $h \leq H$. If $C_h^*(\tau_h, v) = \infty$, then trivially $\bar{C}_h^*(s, v) \leq C_h^*(\tau_h, v)$. Instead, suppose that $C_h^*(\tau_h, v) < \infty$. Then, there must exist a feasible $\pi \in \Pi^D$ satisfying $V_h^\pi(\tau_h) \geq v$. Let $a^* = \pi_h(\tau_h)$. By the policy evaluation equations, we know that,

$$V_h^\pi(\tau_h) = r_h(s, a^*) + \sum_{s'} P_h(s' \mid s, a^*) V_{h+1}^\pi(\tau_h, a^*, s').$$

For each $s' \in \mathcal{S}$, define $v_{s'}^* \stackrel{\text{def}}{=} V_{h+1}^\pi(\tau_h, a^*, s')$ and observe that $v_{s'}^* \in \mathcal{V}_{h+1}(s') \subseteq \mathcal{V}$ by Lemma 3. Thus, we see that $(a^*, \mathbf{v}^*) \in \mathcal{A} \times \mathcal{V}^S$ and $r_h(s, a) + \sum_{s'} P_h(s' \mid s, a) v_{s'} \geq v$, which implies $(a^*, \mathbf{v}^*) \in \bar{\mathcal{A}}_h(s, v)$.

Since $\pi$ satisfies $V_{h+1}^\pi(\tau_h, a^*, s') \geq v_{s'}^*$, it is clear that $C_{h+1}^*(s', v_{s'}^*) \leq C_{h+1}^\pi(\tau_h, a^*, s')$. Thus, the induction hypothesis implies that $\bar{C}_{h+1}^*(s', v_{s'}^*) \leq C_{h+1}^*(s', v_{s'}^*) \leq C_{h+1}^\pi(\tau_h, a^*, s')$. The optimality equations for $\bar{M}$ then imply that,

$$
\begin{aligned}
\bar{C}_h^*(s, v) &= \min_{(a, \mathbf{v}) \in \bar{\mathcal{A}}_h(s, v)} c_h(s, a) + f\left( \left(P_h(s' \mid s, a), \bar{C}_{h+1}^*(s', v_{s'})\right)_{s' \in P_h(s, a)} \right) \\
&\leq c_h(s, a^*) + f\left( \left(P_h(s' \mid s, a^*), \bar{C}_{h+1}^*(s', v_{s'}^*)\right)_{s' \in P_h(s, a^*)} \right) \\
&\leq c_h(s, a^*) + f\left( \left(P_h(s' \mid s, a), C_{h+1}^\pi(\tau_h, a^*, s')\right)_{s' \in P_h(s, a^*)} \right) \\
&= C_h^\pi(\tau_h).
\end{aligned}
$$

The first inequality used the fact that $(a^*, \mathbf{v}^*) \in \bar{\mathcal{A}}_h(s, v)$. The second inequality relied on $f$ being non-decreasing and the induction hypothesis. The final equality used (TR).

Since $\pi$ was an arbitrary feasible policy for the optimization defining $C_h^*(\tau_h, v)$, we see that $\bar{C}_h^*(s, v) \leq C_h^*(\tau_h, v)$. This completes the proof. $\qquad \square$

## C.5   Proof of Lemma 5

*Proof.* We proceed by induction on $h$. Let $(s, v) \in \bar{\mathcal{S}}$ be arbitrary.

**Base Case.** For the base case, we consider $h = H+1$. By definition and assumption, $\bar{C}_{H+1}^\pi(s, v) = \chi_{\{v \leq 0\}} < \infty$. Thus, it must be the case that $v \leq 0$ and so by definition $\bar{V}_{H+1}^\pi(s, v) = 0 \geq v$.

**Inductive Step.** For the inductive step, we consider $h \leq H$. We decompose $\pi_h(s, v) = (a, \mathbf{v})$ where we know $(a, \mathbf{v}) \in \bar{\mathcal{A}}_h(s, v)$ since $\pi$ has finite cost[4]. Moreover, it must be the case that for any $s' \in \mathcal{S}$ with $P_h(s' \mid s, a) > 0$ that $\bar{C}_{h+1}^\pi(s', v_{s'}) < \infty$ otherwise the property that $f$ outputs $\infty$ when inputted an $\infty$ would imply a contradiction:

$$
\begin{aligned}
\bar{C}_h^\pi(s, v) &= c_h(s, a) + f\left( \left(P_h(s' \mid s, a), \bar{C}_{h+1}^\pi(s', v_{s'})\right)_{s' \in P_h(s, a)} \right) \\
&= c_h(s, a) + f(\ldots, \infty, \ldots) \\
&= \infty.
\end{aligned}
$$

Thus, the induction hypothesis implies that $\bar{V}_{h+1}^\pi(s', v_{s'}) \geq v_{s'}$ for any such $s' \in \mathcal{S}$. By the policy evaluation equations, we see that,

$$
\begin{aligned}
\bar{V}_h^\pi(s, v) &= r_h(s, a) + \sum_{s'} P_h(s' \mid s, a) \bar{V}_{h+1}^\pi(s', v_{s'}) \\
&\geq r_h(s, a) + \sum_{s'} P_h(s' \mid s, a) v_{s'} \\
&\geq v.
\end{aligned}
$$

The third line uses the definition of $\bar{\mathcal{A}}_h(s, v)$. This completes the proof. $\qquad \square$

---

[4] By convention, we assume $\min \varnothing = \infty$

## C.6 Proof of Theorem 1

*Proof.* If $\bar{C}_1^*(s_0, v) > B$ for all $v \in \mathcal{V}$, then $C_M^* > B$ since otherwise we would have $\bar{C}_1^*(s_0, v) \leq C_1^*(s_0, v) = C_M^* \leq B$ by Lemma 4. Thus, if Algorithm 1 outputs "infeasible" it is correct.

On the other hand, suppose that there exists some $v \in \mathcal{V}$ for which $\bar{C}_1^*(s_0, v) \leq B$. By standard MDP theory, we know that since $\pi \in \Pi^D$ is a solution to $\bar{M}$, it must satisfy the optimality equations. In particular, $\bar{C}_1^\pi(s_0, v) = \bar{C}_1^*(s_0, v) \leq B$. Since $C_M^\pi = \bar{C}_1^\pi(s_0, v)$[5], we see that there exists a $\pi \in \Pi^D$ for which $C_M^\pi \leq B$ and so $V_M^* > -\infty$.

Since $V_M^*$ is the value of some deterministic policy, Lemma 3 implies that $V_M^* \in \mathcal{V}$. Thus, Lemma 5 implies that $V_1^\pi(s_0, V_M^*) \geq V_M^*$ and $C_1^\pi(s_0, V_M^*) \leq C_1^*(s_0, V_M^*) \leq B$. Consequently, running $\pi$ with initial state $\bar{s}_0 = (s_0, V_M^*)$ is an optimal solution to (CON). In either case, Algorithm 1 is correct. $\qquad\square$

# D    Proofs for Section 4

**Definition 10.** We define the exact partial sum,

$$\sigma_{h,\mathbf{v}}^{s,a}(t, u) \stackrel{\text{def}}{=} u + \sum_{s'=t}^{S} P_h(s' \mid s, a)v_{s'}. \tag{16}$$

**Observation 1.** *We observe that both $\sigma$ and $\hat{\sigma}$ can be computed recursively. Specifically, $\sigma_{h,\mathbf{v}}^{s,a}(S + 1, u) = u$ and $\sigma_{h,\mathbf{v}}^{s,a}(t, u) = \sigma_{h,\mathbf{v}}^{s,a}(t, u + P_h(t \mid s, a)v_t)$. Similarly, $\hat{\sigma}_{h,\mathbf{v}}^{s,a}(S + 1, u) = u$ and $\hat{\sigma}_{h,\mathbf{v}}^{s,a}(t, u) = \sigma_{h,\mathbf{v}}^{s,a}(t, \lfloor u + P_h(t \mid s, a)v_t \rfloor_{\mathcal{G}})$.*

For completeness, and to assist with other arguments, we also prove the exact recursion we presented in Definition 3 is correct using Lemma 6.

**Lemma 6.** *For any $t \in [S + 1]$ and $u \in \mathbb{R}$, we have that,*

$$g_{h,v}^{s,a}(t, u) = \min_{\mathbf{v} \in \mathcal{V}^{S-t+1}} g_{h,\mathbf{v}}^{s,a}(t)$$

$$\text{s.t.} \quad u + \sum_{s'=t}^{S} P_h(s' \mid s, a)v_{s'} \geq v. \tag{17}$$

*Moreover, $\bar{C}_h^*(s, v) = \min_{a \in \mathcal{A}} c_h(s, a) + g_{h,v}^{s,a}(1, r_h(s, a))$.*

## D.1    Proof of Lemma 6

*Proof.* We proceed by induction on $t$.

**Base Case.**    For the base case, we consider $t = S + 1$. Since $\sum_{s'=S+1}^{S} P_h(s' \mid s, a)v_{s'} = 0$ is the empty sum, the condition $u + \sum_{s'=S+1}^{S} P_h(s' \mid s, a)v_{s'} \geq v$ is true iff $u \geq v$. Also, for any $\mathbf{v}$, $g_{h,\mathbf{v}}^{s,a}(S + 1) = 0$ by definition. Thus, the minimum defining $g_{h,\mathbf{v}}^{s,a}(S + 1, u)$ is 0 when $u \geq v$ and is $\infty$ due to infeasibility otherwise. In symbols, $g_{h,v}^{s,a}(S + 1, u) = \chi_{\{u \geq v\}}$ as was to be shown.

---

[5]We can view $\bar{C}$ ($\bar{V}$) as the extension of $C$ ($V$) needed to formally evaluate memory-augmented policies. Since we consider deterministic policies, it is trivial to convert any memory-augmented policy into a history-dependent policy that is defined in the original environment $M$.

**Inductive Step.** For the inductive step, we consider $t \leq S$. We see that $g_{h,v}^{s,a}(t, u)$

$$
= \min_{\substack{\mathbf{v} \in \mathcal{V}^{S-t+1} \\ u + \sum_{s'=t}^{S} P_h(s'|s,a) v_{s'} \geq v}} g_{h,\mathbf{v}}^{s,a}(t)
$$

$$
= \min_{\substack{\mathbf{v} \in \mathcal{V}^{S-t+1} \\ u + \sum_{s'=t}^{S} P_h(s'|s,a) v_{s'} \geq v}} \alpha \left( \beta \left( P_h(t \mid s,a), \bar{C}_{h+1}^*(t, v_t) \right), g_{h,\mathbf{v}}^{s,a}(t+1) \right)
$$

$$
= \min_{v_t \in \mathcal{V}} \min_{\substack{\mathbf{v} \in \mathcal{V}^{S-t} \\ (u + P_h(t|s,a) v_t) + \sum_{s'=t+1}^{S} P_h(s'|s,a) v_{s'} \geq v}} \alpha \left( \beta \left( P_h(t \mid s,a), \bar{C}_{h+1}^*(t, v_t) \right), g_{h,\mathbf{v}}^{s,a}(t+1) \right)
$$

$$
= \min_{v_t \in \mathcal{V}} \alpha \left( \beta \left( P_h(t \mid s,a), \bar{C}_{h+1}^*(t, v_t) \right), \min_{\substack{\mathbf{v} \in \mathcal{V}^{S-t} \\ (u + P_h(t|s,a) v_t) + \sum_{s'=t+1}^{S} P_h(s'|s,a) v_{s'} \geq v}} g_{h,\mathbf{v}}^{s,a}(t+1) \right)
$$

$$
= \min_{v_t \in \mathcal{V}} \alpha \left( \beta \left( P_h(t \mid s,a), \bar{C}_{h+1}^*(t, v_t) \right), g_{h,v}^{s,a}(t+1, u + P_h(t \mid s,a) v_t) \right)
$$

The second lined used (SR). The third line split the optimization into the first decision and the remaining decisions and decomposed the sum in the constraint. The fourth line used the fact that $\alpha$ is a non-decreasing function of both its arguments and the fact that the second optimization only concerns the second argument. The last line used the induction hypothesis.

The observation that $\min_{a \in \mathcal{A}} c_h(s,a) + g_{h,v}^{s,a}(1, r_h(s,a)) = \bar{C}_h^*(s, v)$ then follows from the definition of $\bar{A}_h(s, v)$ and (BU):

$$
\min_{a \in \mathcal{A}} c_h(s,a) + g_{h,v}^{s,a}(1, r_h(s,a)) = \min_{a \in \mathcal{A}} c_h(s,a) + \min_{\substack{\mathbf{v} \in \mathcal{V}^S \\ r_h(s,a) + \sum_{s'} P_h(s'|s,a) v_s \geq v}} g_{h,\mathbf{v}}^{s,a}(1)
$$

$$
= \min_{a \in \mathcal{A}} \min_{\substack{\mathbf{v} \in \mathcal{V}^S \\ r_h(s,a) + \sum_{s'} P_h(s'|s,a) v_s \geq v}} c_h(s,a) + g_{h,\mathbf{v}}^{s,a}(1)
$$

$$
= \min_{(a,\mathbf{v}) \in \bar{A}_h(s,v)} c_h(s,a) + g_{h,\mathbf{v}}^{s,a}(1)
$$

$$
= \bar{C}_h^*(s, v).
$$

$\square$

### D.2 Proof of Lemma 1

*Proof.* We proceed by induction on $t$.

**Base Case.** For the base case, we consider $t = S + 1$. By definition, $\hat{\sigma}_{h,\hat{\mathbf{v}}}^{s,a}(S+1, u) = u$ so the constraint is satisfied iff $u \geq v$. Since for any $\hat{\mathbf{v}}$, $\hat{g}_{h,\hat{\mathbf{v}}}^{s,a}(S+1) = 0$ by definition, the minimum defining $\hat{g}_{h,\hat{\mathbf{v}}}^{s,a}(S+1, u)$ is 0 when $u \geq v$ and is $\infty$ due to infeasibility otherwise. In symbols, $\hat{g}_{h,v}^{s,a}(S+1, u) = \chi_{\{u \geq v\}}$ as was to be shown.

**Inductive Step.** For the inductive step, we consider $t \leq S$. We see that,

$$\hat{g}_{h,v}^{s,a}(t,u) = \min_{\substack{\mathbf{v} \in \mathcal{V}^{S-t+1} \\ \hat{\sigma}_{h,\mathbf{v}}^{s,a}(t,u) \geq v}} g_{h,\mathbf{v}}^{s,a}(t)$$

$$= \min_{\substack{\mathbf{v} \in \mathcal{V}^{S-t+1} \\ \hat{\sigma}_{h,\mathbf{v}}^{s,a}(t,u) \geq v}} \alpha \left( \beta \left( P_h(t \mid s,a), \bar{C}_{h+1}^*(t,v_t) \right), g_{h,\mathbf{v}}^{s,a}(t+1) \right)$$

$$= \min_{v_t \in \mathcal{V}} \min_{\substack{\mathbf{v} \in \mathcal{V}^{S-t} \\ \hat{\sigma}_{h,\mathbf{v}}^{s,a}(t+1, \lfloor u + P_h(t|s,a)v_t \rfloor_{\mathcal{G}}) \geq v}} \alpha \left( \beta \left( P_h(t \mid s,a), \bar{C}_{h+1}^*(t,v_t) \right), g_{h,\mathbf{v}}^{s,a}(t+1) \right)$$

$$= \min_{v_t \in \mathcal{V}} \alpha \left( \beta \left( P_h(t \mid s,a), \bar{C}_{h+1}^*(t,v_t) \right), \min_{\substack{\mathbf{v} \in \mathcal{V}^{S-t} \\ \hat{\sigma}_{h,\mathbf{v}}^{s,a}(t+1, \lfloor u + P_h(t|s,a)v_t \rfloor_{\mathcal{G}}) \geq v}} g_{h,\mathbf{v}}^{s,a}(t+1) \right)$$

$$= \min_{v_t \in \mathcal{V}} \alpha \left( \beta \left( P_h(t \mid s,a), \bar{C}_{h+1}^*(t,v_t) \right), \hat{g}_{h,v}^{s,a}(t+1, \lfloor u + P_h(t \mid s,a)v_t \rfloor_{\mathcal{G}}) \right)$$

The second lined used (SR). The third line split the optimization into the first decision and the remaining decisions and used the recursive definition of $\hat{\sigma}$ in the constraint. The fourth line used the fact that $\alpha$ is a non-decreasing function of both its arguments and the fact that the second optimization only concerns the second argument. The last line used the induction hypothesis. $\qquad \square$

### D.3 Proof of Proposition 4

*Proof.* The runtime guarantee is easily seen since Algorithm 4 consists of nested loops. The fact that it computes an optimal solution for $\bar{M}$ absent rounding or lower bounding follows immediately from Lemma 6. $\qquad \square$

## E Proofs for Section 5

### E.1 Helpful Technical Lemmas (Additive)

The following claims all assume Definition 7.

**Observation 2.** *For any $v \in \mathbb{R}$,*

$$v - \delta \leq \lfloor v \rfloor_{\mathcal{G}} \leq v. \tag{18}$$

**Lemma 7.** *For any $h \in [H]$, $s \in \mathcal{S}$, $a \in \mathcal{A}$, $\mathbf{v} \in \mathbb{R}^S$, $u \in \mathbb{R}$, and $t \in [S+1]$, we have,*

$$\sigma_{h,\mathbf{v}}^{s,a}(t,u) - (S-t+1)\delta \leq \hat{\sigma}_{h,\mathbf{v}}^{s,a}(t,u) \leq \sigma_{h,\mathbf{v}}^{s,a}(t,u). \tag{19}$$

**Lemma 8** (Cost)**.** *For any $h \in [H+1]$ and $(s,v) \in \bar{\mathcal{S}}$, $\hat{C}_h^*(s, \lfloor v \rfloor_{\mathcal{G}}) \leq \bar{C}_h^*(s,v)$.*

**Lemma 9** (Approximation)**.** *Suppose that $\pi \in \Pi^D$. For all $h \in [H+1]$ and $(s,\hat{v}) \in \hat{\mathcal{S}}$, if $\hat{C}_h^\pi(s,\hat{v}) < \infty$, then $\hat{V}_h^\pi(s,\hat{v}) \geq \hat{v} - \delta(S+1)(H-h+1)$.*

### E.2 Helpful Technical Lemmas (Relative)

The following claims all assume Definition 8.

**Observation 3.** *For any $v \in \mathbb{R}$,*

$$v(1-\delta) \leq \lfloor v \rfloor_{\mathcal{G}} \leq v. \tag{20}$$

**Lemma 10.** *For any $h \in [H]$, $s \in \mathcal{S}$, $a \in \mathcal{A}$, $\mathbf{v} \in \mathbb{R}_{\geq 0}^S$, $u \in \mathbb{R}_{\geq 0}$, and $t \in [S+1]$, we have,*

$$\sigma_{h,\mathbf{v}}^{s,a}(t,u)(1-\delta)^{S-t+1} \leq \hat{\sigma}_{h,\mathbf{v}}^{s,a}(t,u) \leq \sigma_{h,\mathbf{v}}^{s,a}(t,u). \tag{21}$$

**Lemma 11** (Cost)**.** *Suppose all rewards are non-negative. For any $h \in [H+1]$ and $(s,v) \in \bar{\mathcal{S}}$, $\hat{C}_h^*(s, \lfloor v \rfloor_{\mathcal{G}}) \leq \bar{C}_h^*(s,v)$.*

**Lemma 12** (Approximation)**.** *Suppose all rewards are non-negative and $\pi \in \Pi^D$. For all $h \in [H+1]$ and $(s,\hat{v}) \in \hat{\mathcal{S}}$, if $\hat{C}_h^\pi(s,\hat{v}) < \infty$, then $\hat{V}_h^\pi(s,\hat{v}) \geq \hat{v}(1-\delta)^{(S+1)(H-h+1)}$.*

### E.3 Proof of Observation 2

*Proof.* Using properties of the floor function, we can infer that,

$$\lfloor v \rfloor_{\mathcal{G}} = \left\lfloor \frac{v}{\delta} \right\rfloor \delta \leq \frac{v}{\delta} \delta = v,$$

and,

$$\lfloor v \rfloor_{\mathcal{G}} = \left\lfloor \frac{v}{\delta} \right\rfloor \delta \geq (\left\lceil \frac{v}{\delta} \right\rceil - 1)\delta = \left\lceil \frac{v}{\delta} \right\rceil \delta - \delta \geq v - \delta.$$

$\square$

### E.4 Proof of Lemma 7

*Proof.* We proceed by induction on $t$.

**Base Case.** For the base case, we consider $t = S + 1$. By definition, we have $\hat{\sigma}_{h,\mathbf{v}}^{s,a}(S + 1, u) = u = \sigma_{h,\mathbf{v}}^{s,a}(S + 1, u)$.

**Inductive Step.** For the inductive step, we consider $t \leq S$. We first see that,

$$
\begin{aligned}
\hat{\sigma}_{h,\hat{\mathbf{v}}}^{s,a}(t, u) &= \hat{\sigma}_{h,\hat{\mathbf{v}}}^{s,a}(t + 1, \lfloor u + P_h(t \mid s, a)\hat{v}_t \rfloor_{\mathcal{G}}) \\
&\leq \sigma_{h,\hat{\mathbf{v}}}^{s,a}(t + 1, \lfloor u + P_h(t \mid s, a)\hat{v}_t \rfloor_{\mathcal{G}}) \\
&= \lfloor u + P_h(t \mid s, a)\hat{v}_t \rfloor_{\mathcal{G}} + \sum_{s'=t+1}^{S} P_h(s' \mid s, a)\hat{v}_t \\
&\leq u + \sum_{s'=t}^{S} P_h(s' \mid s, a)\hat{v}_t \\
&= \sigma_{h,\hat{\mathbf{v}}}^{s,a}(t, u).
\end{aligned}
$$

The first inequality used the induction hypothesis and the second inequality used the fact that $\lfloor x \rfloor_{\mathcal{G}} \leq x$.

We also see that,

$$
\begin{aligned}
\hat{\sigma}_{h,\hat{\mathbf{v}}}^{s,a}(t, u) &= \hat{\sigma}_{h,\hat{\mathbf{v}}}^{s,a}(t + 1, \lfloor u + P_h(t \mid s, a)\hat{v}_t \rfloor_{\mathcal{G}}) \\
&\geq \sigma_{h,\hat{\mathbf{v}}}^{s,a}(t + 1, \lfloor u + P_h(t \mid s, a)\hat{v}_t \rfloor_{\mathcal{G}}) - \delta(S - t) \\
&= \lfloor u + P_h(t \mid s, a)\hat{v}_t \rfloor_{\mathcal{G}} + \sum_{s'=t+1}^{S} P_h(s' \mid s, a)\hat{v}_t - \delta(S - t) \\
&\geq u + \sum_{s'=t}^{S} P_h(s' \mid s, a)\hat{v}_t - \delta(S - t + 1) \\
&= \sigma_{h,\hat{\mathbf{v}}}^{s,a}(t, u) - \delta(S - t + 1).
\end{aligned}
$$

The first inequality used the induction hypothesis and the second inequality used the fact that $\lfloor x \rfloor_{\mathcal{G}} \geq x - \delta$. $\square$

### E.5 Proof of Lemma 8

*Proof.* We proceed by induction on $h$. Let $(s, v) \in \bar{\mathcal{S}}$ be arbitrary.

**Base Case.** For the base case, we consider $h = H + 1$. Since $\lfloor v \rfloor_{\mathcal{G}} \leq v$, we immediately see,

$$\hat{C}_{H+1}^*(s, \lfloor v \rfloor_{\mathcal{G}}) = \chi_{\{\lfloor v \rfloor_{\mathcal{G}} \leq 0\}} \leq \chi_{\{v \leq 0\}} = \bar{C}_{H+1}^*(s, v).$$

**Inductive Step.** For the inductive step, we consider $h \leq H$. If $\bar{C}_h^*(s,v) = \infty$, then trivially $\hat{C}_h^*(s, \lfloor v \rfloor_\mathcal{G}) \leq \bar{C}_h^*(s,v)$. Instead, suppose that $\bar{C}_h^*(s,v) < \infty$. Let $\pi$ be a solution to the optimality equations for $\bar{M}$ so that $\bar{C}_h^\pi(s,v) = \bar{C}_h^*(s,v) < \infty$. Since $\bar{C}_h^*(s,v) < \infty$, we know that $(a^*, \mathbf{v}^*) = \pi_h(s,v) \in \bar{A}_h(s,v)$. By the definition of $\bar{A}_h(s,v)$, we know that,

$$\sigma_{h,\mathbf{v}^*}^{s,a^*}(1, r_h(s,a^*)) = r_h(s,a^*) + \sum_{s'} P_h(s' \mid s, a^*) v_{s'}^* \geq v \geq \lfloor v \rfloor_\mathcal{G}.$$

For each $s' \in \mathcal{S}$, define $\hat{v}_{s'}^* \overset{\text{def}}{=} \lfloor v_{s'}^* \rfloor_\mathcal{G}$ and recall that $v_{s'}^* \in \mathcal{V}$. We first observe that,

$$\sigma_{h,\hat{\mathbf{v}}^*}^{s,a^*}(1, r_h(s,a^*)) = r_h(s,a^*) + \sum_{s'} P_h(s' \mid s, a) \lfloor v_{s'} \rfloor_\mathcal{G}$$

$$\geq r_h(s,a^*) + \sum_{s'} P_h(s' \mid s, a)(v_{s'} - \delta)$$

$$= r_h(s,a^*) + \sum_{s'} P_h(s' \mid s, a) v_{s'} - \delta$$

$$= \sigma_{h,\mathbf{v}^*}^{s,a^*}(1, r_h(s,a^*)) - \delta.$$

Then by Lemma 7,

$$\hat{\sigma}_{h,\hat{\mathbf{v}}^*}^{s,a^*}(1, r_h(s,a^*)) \geq \sigma_{h,\hat{\mathbf{v}}^*}^{s,a^*}(1, r_h(s,a^*)) - \delta S$$

$$\geq \sigma_{h,\mathbf{v}^*}^{s,a^*}(1, r_h(s,a^*)) - \delta(S+1)$$

$$\geq \lfloor v \rfloor_\mathcal{G} - \delta(S+1)$$

$$= \kappa(\lfloor v \rfloor_\mathcal{G}).$$

Thus, $(a^*, \hat{\mathbf{v}}^*) \in \hat{A}_h(s, \lfloor v \rfloor_\mathcal{G})$.

Since $v_{s'}^* \in \mathcal{V}$, the induction hypothesis implies that $\hat{C}_{h+1}^*(s', \hat{v}_{s'}^*) \leq \bar{C}_{h+1}^*(s', v_{s'}^*) = \bar{C}_{h+1}^\pi(s', v_{s'}^*)$. The optimality equations for $\hat{M}$ then imply that,

$$\hat{C}_h^*(s, \lfloor v \rfloor_\mathcal{G}) = \min_{(a,\hat{\mathbf{v}}) \in \hat{A}_h(s,v)} c_h(s,a) + f\left(\left(P_h(s' \mid s, a), \hat{C}_{h+1}^*(s', \hat{v}_{s'})\right)_{s' \in P_h(s,a)}\right)$$

$$\leq c_h(s,a^*) + f\left(\left(P_h(s' \mid s, a^*), \hat{C}_{h+1}^*(s', \hat{v}_{s'}^*)\right)_{s' \in P_h(s,a^*)}\right)$$

$$\leq c_h(s,a^*) + f\left(\left(P_h(s' \mid s, a), \bar{C}_{h+1}^\pi(s', v_{s'}^*)\right)_{s' \in P_h(s,a^*)}\right)$$

$$= \bar{C}_h^\pi(s,v)$$

$$= \bar{C}_h^*(s,v).$$

The first inequality used the fact that $(a^*, \mathbf{v}^*) \in \hat{A}_h(s,v)$. The second inequality relied on $f$ being non-decreasing and the induction hypothesis. The penultimate equality used (TR). This completes the proof. $\qquad\square$

## E.6 Proof of Lemma 9

*Proof.* We proceed by induction on $h$. Let $(s, \hat{v}) \in \hat{\mathcal{S}}$ be arbitrary.

**Base Case.** For the base case, we consider $h = H+1$. By definition and assumption, $\hat{C}_{H+1}^\pi(s, \hat{v}) = \chi_{\{\hat{v} \leq 0\}} < \infty$. Thus, it must be the case that $\hat{v} \leq 0$ and so by definition $\hat{V}_{H+1}^\pi(s, \hat{v}) = 0 \geq \hat{v}$.

**Inductive Step.** For the inductive step, we consider $h \leq H$. As in the proof of Lemma 5, we know that $\pi_h(s,v) = (a, \hat{\mathbf{v}}) \in \hat{A}_h(s, \hat{v})$ and for any $s' \in \mathcal{S}$ with $P_h(s' \mid s, a) > 0$ that $\hat{C}_{h+1}^\pi(s', v_{s'}) < \infty$. Thus, the induction hypothesis implies that $\hat{V}_{h+1}^\pi(s', \hat{v}_{s'}) \geq \hat{v}_{s'} - \delta(S+1)(H-h)$ for any such

$s' \in \mathcal{S}$. By the policy evaluation equations, we see that,

$$\hat{V}_h^\pi(s, \hat{v}) = r_h(s, a) + \sum_{s'} P_h(s' \mid s, a) \hat{V}_{h+1}^\pi(s', \hat{v}_{s'})$$

$$\geq r_h(s, a) + \sum_{s'} P_h(s' \mid s, a) \hat{v}_{s'} - \delta(S+1)(H-h)$$

$$= \sigma_{h,\hat{\mathbf{v}}}^{s,a}(1, r_h(s, a)) - \delta(S+1)(H-h)$$

$$\geq \hat{\sigma}_{h,\hat{\mathbf{v}}}^{s,a}(1, r_h(s, a)) - \delta(S+1)(H-h)$$

$$\geq \hat{v} - \delta(S+1) - \delta(S+1)(H-h)$$

$$= \hat{v} - \delta(S+1)(H-h+1).$$

The first inequality used the induction hypothesis. The second inequality used Lemma 7. The third inequality used the fact that by definition of $\hat{\mathcal{A}}_h(s, \hat{v})$ and $\kappa$, $\hat{\sigma}_{h,\hat{\mathbf{v}}}^{s,a}(1, r_h(s, a)) \geq \kappa(\hat{v}) = \hat{v} - \delta(S+1)$. This completes the proof. $\qquad\square$

### E.7 Proof of Theorem 2

*Proof.*

**Correctness.** If $\hat{C}_1^*(s_0, v) > B$ for all $\hat{v} \in \hat{\mathcal{V}}$, then $C_M^* > B$ since otherwise we would have $\hat{C}_1^*(s_0, \lfloor v \rfloor_{\mathcal{G}}) \leq \bar{C}_1^*(s_0, v) \leq C_M^* \leq B$ by Lemma 8. Thus, if Algorithm 5 outputs "infeasible" it is correct.

On the other hand, suppose that there exists some $\hat{v} \in \hat{\mathcal{V}}$ for which $\hat{C}_1^*(s_0, \hat{v}) \leq B$. By standard MDP theory, we know that since $\pi \in \Pi^D$ is a solution to $\hat{M}$, it must satisfy the optimality equations. In particular, $\hat{C}_1^\pi(s_0, \hat{v}) = \hat{C}_1^*(s_0, v) \leq B$. As in the proof of Theorem 1, since $C_M^\pi = \hat{C}_1^\pi(s_0, \hat{v})$, we see that there exists a $\pi \in \Pi^D$ for which $C_M^\pi \leq B$ and so $V_M^* > -\infty$.

Since $V_M^*$ is the value of some deterministic policy, Lemma 3 implies that $V_M^* \in \mathcal{V}$. Thus, Lemma 9 implies that $\hat{V}_1^\pi(s_0, \lfloor V_M^* \rfloor_{\mathcal{G}}) \geq \lfloor V_M^* \rfloor_{\mathcal{G}} - \delta(S+1)H \geq V_M^* - \delta(1 + (S+1)H) = V_M^* - \epsilon$ and $\hat{C}_1^\pi(s_0, V_M^*) \leq C_1^*(s_0, V_M^*) \leq B$. Consequently, running $\pi$ with initial state $\bar{s}_0 = (s_0, \lfloor V_M^* \rfloor_{\mathcal{G}})$ is an optimal solution to (CON). In either case, Algorithm 5 is correct.

**Complexity.** For the complexity claim, we observe that the running time of Algorithm 5 is $O(HS^2A|\hat{\mathcal{V}}|^2|\hat{\mathcal{U}}|)$. To bound $|\hat{\mathcal{V}}|$, we observe that the number of integer multiples of $\delta$ required to capture the range $[-Hr_{max}, Hr_{max}]$ is at most $O(\frac{Hr_{max}}{\delta}) = O(H^2Sr_{max}/\epsilon)$ by definition of $\delta$. Moreover, $|\hat{\mathcal{U}}| = O(|\hat{\mathcal{V}}| + S) = O(|\hat{\mathcal{V}}|)$ for sufficiently large $\frac{r_{max}}{\epsilon}$.

In particular, we see that the range of the rounded sums defining $\hat{\mathcal{U}}$ is at widest $[-2Hr_{max} - \delta S, 2Hr_{max}]$ since for any $t + 1$ the rounded input is,

$$\left\lfloor \lfloor r_h(s, a) + P_h(1 \mid s, a)\hat{\mathbf{v}}_1 \rfloor_{\mathcal{G}} + \ldots + P_h(t \mid s, a)\hat{\mathbf{v}}_t \right\rfloor_{\mathcal{G}} \leq r_h(s, a) + \sum_{s'=1}^{t} P_h(s' \mid s, a)\hat{\mathbf{v}}_{s'},$$

which is at most $2Hr_{max}$, and,

$$\left\lfloor \lfloor r_h(s, a) + P_h(1 \mid s, a)\hat{\mathbf{v}}_1 \rfloor_{\mathcal{G}} + \ldots + P_h(t \mid s, a)\hat{\mathbf{v}}_t \right\rfloor_{\mathcal{G}} \geq r_h(s, a) + \sum_{s'=1}^{t} P_h(s' \mid s, a)\hat{\mathbf{v}}_{s'} - \delta t,$$

which is at least $-2Hr_{max} - \delta S$. Overall, we see that $O(|\hat{\mathcal{V}}|^2|\hat{\mathcal{U}}|) = O(|\hat{\mathcal{V}}|^3) = O(H^6S^3r_{max}^3/\epsilon^3)$ implying that the total run time is $O(H^7S^5Ar_{max}^3/\epsilon^3)$ as claimed. $\qquad\square$

### E.8 Proof of Observation 3

*Proof.* Using properties of the floor function, we can infer that,

$$\lfloor v \rfloor_{\mathcal{G}} = v^{min} \left( \frac{1}{1-\delta} \right)^{\left\lfloor \log_{\frac{1}{1-\delta}} \frac{v}{v^{min}} \right\rfloor} \leq v^{min} \left( \frac{1}{1-\delta} \right)^{\log_{\frac{1}{1-\delta}} \frac{v}{v^{min}}} = \frac{v}{v^{min}} v^{min} = v,$$

and,

$$\lfloor v \rfloor_{\mathcal{G}} = v^{min} \left( \frac{1}{1-\delta} \right)^{\left\lfloor \log_{\frac{1}{1-\delta}} \frac{v}{v^{min}} \right\rfloor} \geq v^{min} \left( \frac{1}{1-\delta} \right)^{\log_{\frac{1}{1-\delta}} \frac{v}{v^{min}} - 1} = v(1-\delta).$$

□

## E.9 Proof of Lemma 10

*Proof.* We proceed by induction on $t$.

**Base Case.** For the base case, we consider $t = S + 1$. By definition, we have $\hat{\sigma}_{h,\mathbf{v}}^{s,a}(S+1, u) = u = \sigma_{h,\mathbf{v}}^{s,a}(S+1, u)$.

**Inductive Step.** For the inductive step, we consider $t \leq S$. We first see that,

$$\begin{aligned}
\hat{\sigma}_{h,\hat{\mathbf{v}}}^{s,a}(t, u) &= \hat{\sigma}_{h,\hat{\mathbf{v}}}^{s,a}(t+1, \lfloor u + P_h(t \mid s, a)\hat{v}_t \rfloor_{\mathcal{G}}) \\
&\leq \sigma_{h,\hat{\mathbf{v}}}^{s,a}(t+1, \lfloor u + P_h(t \mid s, a)\hat{v}_t \rfloor_{\mathcal{G}}) \\
&= \lfloor u + P_h(t \mid s, a)\hat{v}_t \rfloor_{\mathcal{G}} + \sum_{s'=t+1}^{S} P_h(s' \mid s, a)\hat{v}_t \\
&\leq u + \sum_{s'=t}^{S} P_h(s' \mid s, a)\hat{v}_t \\
&= \sigma_{h,\hat{\mathbf{v}}}^{s,a}(t, u).
\end{aligned}$$

The first inequality used the induction hypothesis and the second inequality used the fact that $\lfloor x \rfloor_{\mathcal{G}} \leq x$.

We also see that,

$$\begin{aligned}
\hat{\sigma}_{h,\hat{\mathbf{v}}}^{s,a}(t, u) &= \hat{\sigma}_{h,\hat{\mathbf{v}}}^{s,a}\left(t+1, \lfloor u + P_h(t \mid s, a)\hat{v}_t \rfloor_{\mathcal{G}}\right) \\
&\geq \sigma_{h,\hat{\mathbf{v}}}^{s,a}\left(t+1, \lfloor u + P_h(t \mid s, a)\hat{v}_t \rfloor_{\mathcal{G}}\right)(1-\delta)^{S-t} \\
&= \left( \lfloor u + P_h(t \mid s, a)\hat{v}_t \rfloor_{\mathcal{G}} + \sum_{s'=t+1}^{S} P_h(s' \mid s, a)\hat{v}_t \right)(1-\delta)^{S-t} \\
&\geq \left( (1-\delta)u + (1-\delta)\sum_{s'=t}^{S} P_h(s' \mid s, a)\hat{v}_t \right)(1-\delta)^{S-t} \\
&= \sigma_{h,\hat{\mathbf{v}}}^{s,a}(t, u)(1-\delta)^{S-t+1}.
\end{aligned}$$

The first inequality used the induction hypothesis and the second inequality used the fact that $\lfloor x \rfloor_{\mathcal{G}} \geq x - \delta$ and the fact that all rewards and values are non-negative allowing us to add a $(1-\delta)$-factor to the other value demands. □

## E.10 Proof of Lemma 11

*Proof.* We proceed by induction on $h$. Let $(s, v) \in \bar{\mathcal{S}}$ be arbitrary.

**Base Case.** For the base case, we consider $h = H + 1$. Since $\lfloor v \rfloor_{\mathcal{G}} \leq v$, we immediately see,

$$\hat{C}_{H+1}^*(s, \lfloor v \rfloor_{\mathcal{G}}) = \chi_{\{\lfloor v \rfloor_{\mathcal{G}} \leq 0\}} \leq \chi_{\{v \leq 0\}} = \bar{C}_{H+1}^*(s, v).$$

**Inductive Step.** For the inductive step, we consider $h \leq H$. If $\bar{C}_h^*(s, v) = \infty$, then trivially $\hat{C}_h^*(s, \lfloor v \rfloor_{\mathcal{G}}) \leq \bar{C}_h^*(s, v)$. Instead, suppose that $\bar{C}_h^*(s, v) < \infty$. Let $\pi$ be a solution to the optimality

equations for $\bar{M}$ so that $\bar{C}_h^\pi(s,v) = \bar{C}_h^*(s,v) < \infty$. Since $\bar{C}_h^*(s,v) < \infty$, we know that $(a^*, \mathbf{v}^*) = \pi_h(s,v) \in \bar{\mathcal{A}}_h(s,v)$. By the definition of $\bar{\mathcal{A}}_h(s,v)$, we know that,

$$\sigma_{h,\mathbf{v}^*}^{s,a^*}(1, r_h(s,a^*)) = r_h(s,a^*) + \sum_{s'} P_h(s' \mid s, a^*) v_{s'}^* \geq v \geq \lfloor v \rfloor_{\mathcal{G}}.$$

For each $s' \in \mathcal{S}$, define $\hat{v}_{s'}^* \overset{\text{def}}{=} \lfloor v_{s'}^* \rfloor_{\mathcal{G}}$ and recall that $v_{s'}^* \in \mathcal{V}$. We first observe that,

$$\sigma_{h,\hat{\mathbf{v}}^*}^{s,a^*}(1, r_h(s,a^*)) = r_h(s,a^*) + \sum_{s'} P_h(s' \mid s, a) \lfloor v_{s'} \rfloor_{\mathcal{G}}$$

$$\geq r_h(s,a^*) + \sum_{s'} P_h(s' \mid s, a) v_{s'}(1 - \delta)$$

$$\geq \left( r_h(s,a^*) + \sum_{s'} P_h(s' \mid s, a) v_{s'} \right)(1 - \delta)$$

$$= \sigma_{h,\mathbf{v}^*}^{s,a^*}(1, r_h(s,a^*))(1 - \delta).$$

The second inequality used the fact that all rewards are non-negative. Then by Lemma 10,

$$\hat{\sigma}_{h,\hat{\mathbf{v}}^*}^{s,a^*}(1, r_h(s,a^*)) \geq \sigma_{h,\hat{\mathbf{v}}^*}^{s,a^*}(1, r_h(s,a^*))(1 - \delta)^S$$

$$\geq \sigma_{h,\mathbf{v}^*}^{s,a^*}(1, r_h(s,a^*))(1 - \delta)^{S+1}$$

$$\geq \lfloor v \rfloor_{\mathcal{G}}(1 - \delta)^{S+1}$$

$$= \kappa(\lfloor v \rfloor_{\mathcal{G}}).$$

Thus, $(a^*, \hat{\mathbf{v}}^*) \in \hat{\mathcal{A}}_h(s, \lfloor v \rfloor_{\mathcal{G}})$.

Since $v_{s'}^* \in \mathcal{V}$, the induction hypothesis implies that $\hat{C}_{h+1}^*(s', \hat{v}_{s'}^*) \leq \bar{C}_{h+1}^*(s', v_{s'}^*) = \bar{C}_{h+1}^\pi(s', v_{s'}^*)$. The optimality equations for $\hat{M}$ then imply that,

$$\hat{C}_h^*(s, \lfloor v \rfloor_{\mathcal{G}}) = \min_{(a,\hat{\mathbf{v}}) \in \hat{\mathcal{A}}_h(s,v)} c_h(s,a) + f\left( \left( P_h(s' \mid s, a), \hat{C}_{h+1}^*(s', \hat{v}_{s'}) \right)_{s' \in P_h(s,a)} \right)$$

$$\leq c_h(s,a^*) + f\left( \left( P_h(s' \mid s, a^*), \hat{C}_{h+1}^*(s', \hat{v}_{s'}^*) \right)_{s' \in P_h(s,a^*)} \right)$$

$$\leq c_h(s,a^*) + f\left( \left( P_h(s' \mid s, a), \bar{C}_{h+1}^\pi(s', v_{s'}^*) \right)_{s' \in P_h(s,a^*)} \right)$$

$$= \bar{C}_h^\pi(s,v)$$

$$= \bar{C}_h^*(s,v).$$

The first inequality used the fact that $(a^*, \mathbf{v}^*) \in \hat{\mathcal{A}}_h(s,v)$. The second inequality relied on $f$ being non-decreasing and the induction hypothesis. The penultimate equality used (TR).

This completes the proof. □

### E.11 Proof of Lemma 12

*Proof.* We proceed by induction on $h$. Let $(s, \hat{v}) \in \hat{\mathcal{S}}$ be arbitrary.

**Base Case.** For the base case, we consider $h = H+1$. By definition and assumption, $\hat{C}_{H+1}^\pi(s, \hat{v}) = \chi_{\{\hat{v} \leq 0\}} < \infty$. Thus, it must be the case that $\hat{v} \leq 0$ and so by definition $\hat{V}_{H+1}^\pi(s, \hat{v}) = 0 \geq \hat{v}$.

**Inductive Step.** For the inductive step, we consider $h \leq H$. As in the proof of Lemma 5, we know that $\pi_h(s,v) = (a, \hat{v}) \in \hat{\mathcal{A}}_h(s, \hat{v})$ and for any $s' \in \mathcal{S}$ with $P_h(s' \mid s, a) > 0$ that $\hat{C}_{h+1}^\pi(s', v_{s'}) < \infty$. Thus, the induction hypothesis implies that $\hat{V}_{h+1}^\pi(s', \hat{v}_{s'}) \geq \hat{v}_{s'}(1 - \delta)^{(S+1)(H-h)}$ for any such

$s' \in \mathcal{S}$. By the policy evaluation equations, we see that,

$$\hat{V}_h^\pi(s, \hat{v}) = r_h(s, a) + \sum_{s'} P_h(s' \mid s, a)\hat{V}_{h+1}^\pi(s', \hat{v}_{s'})$$

$$\geq r_h(s, a) + \sum_{s'} P_h(s' \mid s, a)\hat{v}_{s'}(1 - \delta)^{(S+1)(H-h)}$$

$$\geq \sigma_{h,\hat{\mathbf{v}}}^{s,a}(1, r_h(s, a))(1 - \delta)^{(S+1)(H-h)}$$

$$\geq \hat{\sigma}_{h,\hat{\mathbf{v}}}^{s,a}(1, r_h(s, a))(1 - \delta)^{(S+1)(H-h)}$$

$$\geq \hat{v}(1 - \delta)^{S+1}(1 - \delta)^{(S+1)(H-h)}$$

$$= \hat{v}(1 - \delta)^{(S+1)(H-h+1)}.$$

The first inequality used the induction hypothesis. The second inequality used the fact that the rewards are non-negative. The third inequality used Lemma 10. The fourth inequality used the fact that by definition of $\hat{\mathcal{A}}_h(s, \hat{v})$ and $\kappa$, $\hat{\sigma}_{h,\hat{\mathbf{v}}}^{s,a}(1, r_h(s, a)) \geq \kappa(\hat{v}) = \hat{v}(1 - \delta)^{S+1}$.

This completes the proof. $\qquad\square$

### E.12 Proof of Theorem 3

*Proof.*

**Correctness.** If $\hat{C}_1^*(s_0, v) > B$ for all $\hat{v} \in \hat{\mathcal{V}}$, then $C_M^* > B$ since otherwise we would have $\hat{C}_1^*(s_0, \lfloor v \rfloor_{\mathcal{G}}) \leq \bar{C}_1^*(s_0, v) \leq C_M^* \leq B$ by Lemma 11. Thus, if Algorithm 5 outputs "infeasible" it is correct.

On the other hand, suppose that there exists some $\hat{v} \in \hat{\mathcal{V}}$ for which $\hat{C}_1^*(s_0, \hat{v}) \leq B$. By standard MDP theory, we know that since $\pi \in \Pi^D$ is a solution to $\hat{M}$, it must satisfy the optimality equations. In particular, $\hat{C}_1^\pi(s_0, \hat{v}) = \hat{C}_1^*(s_0, v) \leq B$. As in the proof of Theorem 1, since $C_M^\pi = \hat{C}_1^\pi(s_0, \hat{v})$, we see that there exists a $\pi \in \Pi^D$ for which $C_M^\pi \leq B$ and so $V_M^* > -\infty$.

Since $V_M^*$ is the value of some deterministic policy, Lemma 3 implies that $V_M^* \in \mathcal{V}$. Thus, Lemma 12 implies that $\hat{V}_1^\pi(s_0, \lfloor V_M^* \rfloor_{\mathcal{G}}) \geq \lfloor V_M^* \rfloor_{\mathcal{G}} (1 - \delta)^{(S+1)H} \geq V_M^*(1 - \delta)^{(S+1)H+1} = V_M^*(1 - \frac{\epsilon}{(S+1)H+1})^{(S+1)H+1} \geq V_M^*(1 - \epsilon)$ and $\hat{C}_1^\pi(s_0, V_M^*) \leq C_1^*(s_0, V_M^*) \leq B$. Consequently, running $\pi$ with initial state $\bar{s}_0 = (s_0, \lfloor V_M^* \rfloor_{\mathcal{G}})$ is an optimal solution to (CON). In either case, Algorithm 5 is correct.

**Complexity.** For the complexity claim, we observe that the running time of Algorithm 5 is $O(HS^2A|\hat{\mathcal{V}}|^2|\hat{\mathcal{U}}|)$. To bound $|\hat{\mathcal{V}}|$, we observe that the number of $vmin$-scaled powers of $1/(1 - \delta)$ required to capture the range $[0, Hr_{max}]$ is at most one plus the largest power needed, which is

$$O(\log_{1/(1-\delta)}(\frac{Hr_{max}}{v_{min}})) = O(\log(\frac{Hr_{max}}{v_{min}})/\log(1/(1 - \delta)))$$

$$= O(\log(\frac{Hr_{max}}{v_{min}})/\delta)$$

$$= O(\log(HS\frac{Hr_{max}}{p_{min}^H r_{min}})/\epsilon)$$

$$= O(H^2S\log(\frac{r_{max}}{p_{min}r_{min}})/\epsilon),$$

by definition of $\delta$ and the fact that $\log(\frac{1}{1-\delta}) = -\log(1 - \delta) \geq -\log(e^{-\delta}) = \delta$. Moreover, $|\hat{\mathcal{U}}| = O(|\hat{\mathcal{V}}|)$.

We see that the range of the rounded sums is at widest $[0, 2Hr_{max}]$ since for any $t + 1$ rounding non-negative sums is at least 0 and,

$$\lfloor \lfloor r_h(s, a) + P_h(1 \mid s, a)\hat{\mathbf{v}}_1 \rfloor_{\mathcal{G}} + \ldots + P_h(t \mid s, a)\hat{\mathbf{v}}_t \rfloor_{\mathcal{G}} \leq r_h(s, a) + \sum_{s'=1}^{t} P_h(s' \mid s, a)\hat{\mathbf{v}}_{s'},$$

which is at most $2Hr_{max}$. Then, the same analysis from before shows that the number of scaled powers of $1/(1-\delta)$ needed to cover this interval is $O(|\hat{\mathcal{V}}|)$. Thus, we see that $O(|\hat{\mathcal{V}}|^2|\hat{\mathcal{U}}|) = O(|\hat{\mathcal{V}}|^3) = O(H^6 S^3 \log(\frac{r_{max}}{p_{min}r_{min}})^3/\epsilon^3)$ implying that the total run time is $O(H^7 S^5 A \log(\frac{r_{max}}{p_{min}r_{min}})^3/\epsilon^3)$ as claimed. $\square$

## F   Extensions

### F.1   Stochastic Costs

Suppose each cost $c_h(s,a)$ is replaced with a cost distribution $C_h(s,a)$. Here, we consider finitely supported cost distributions whose supports are at most $m \in \mathbb{N}$. Then, instead of the agent occurring cost $c_h(s,a)$ upon taking action $a$ in state $s$ at time $h$, the agent occurs a random cost $c_h \sim C_h(s,a)$. Generally, this necessitates histories be cost dependent, and so the policy evaluation equations become,

$$V_h^\pi(\tau_h) = r_h(s,a) + \sum_{c',s'} C_h(c' \mid s,a) P_h(s' \mid s,a) V_{h+1}^\pi(\tau_h, a, c', s'). \tag{CPE}$$

**Cover MDP.**   This implicitly changes the definition of $\mathcal{V}$ since the histories considered in the definition must now include cost history. Since the cost distributions are finitely supported, $\mathcal{V}$ remains a finite set. The main difference for $\bar{M}$ is that the future value demands must depend on both the immediate cost and the next state. This slightly changes the action space:

$$\bar{\mathcal{A}}_h(s,v) \stackrel{\text{def}}{=} \left\{ (a,\mathbf{v}) \in \mathcal{A} \times \mathcal{V}^{m \times S} \mid r_h(s,a) + \sum_{c',s'} C_h(c' \mid s,a) P_h(s' \mid s,a) v_{c',s'} \geq v \right\}.$$

**Bellman Updates.**   In order to solve $\bar{M}$ using Algorithm 4, we must extend the definition of TSR to also be recursive in the immediate costs. The key difference of the *TSRC* condition is that $g$'s recursion is now two dimensional.

**Definition 11** (TSRC).   We call a criterion $C$ *time-space-cost-recursive* (TSRC) if $C_M^\pi = C_1^\pi(s_0)$ where $C_{H+1}^\pi(\cdot) = \mathbf{0}$ and for any $h \in [H]$ and $\tau_h \in \mathcal{H}_h$ letting $s = s_h(\tau_h)$ and $a = \pi_h(\tau_h)$,

$$C_h^\pi(\tau_h) = c_h(s,a) + f\left( \left( C_h(c' \mid s,a), P_h(s' \mid s,a), C_{h+1}^\pi(\tau_h, a, c', s') \right)_{c',s'} \right). \tag{22}$$

In the above, $c' \in C_h(s,a)$ and $s' \in P_h(s,a)$. We now require that $f$ be computable in $O(mS)$ time. We also require that the $f$ term above is equal to $g_h^{\tau_h,a}(1,1)$, where, $g_h^{\tau_h,a}(m+1,1) = 0$, $g_h^{\tau_h,a}(k, S+1) = g_h^{\tau_h,a}(k+1,1)$, and,

$$g_h^{\tau_h,a}(k,t) = \alpha\left( \beta\left( C_h(c_k \mid s,a), P_h(t \mid s,a), C_{h+1}^\pi(\tau_h,a,t) \right), g_h^{\tau_h,a}(k,t+1) \right). \tag{23}$$

In the above, we assume $c_k$ is the $k$th supported cost of $C_h(s,a)$. Again, both $\alpha, \beta$ can be computed in $O(1)$ time, but now $\alpha(\beta(y,\cdot),x) = x$ whenever $0 \in y$.

Our examples from before also carry over to the stochastic cost setting.

**Proposition 5** (TSCR examples).   *The following classical constraints can be modeled by a TSCR cost constraint.*

1. (Expectation Constraints) *We capture these constraints by defining* $\alpha(x,y) \stackrel{\text{def}}{=} x+y$ *and* $\beta(x,y,z) \stackrel{\text{def}}{=} xyz$.

2. (Almost Sure Constraints) *We capture these constraints by defining* $\alpha(x,y) \stackrel{\text{def}}{=} \max(x,y)$ *and* $\beta(x,y,z) \stackrel{\text{def}}{=} [x > 0 \wedge y > 0]z$.

3. (Anytime Constraints) *We capture these constraints by defining* $\alpha(x,y) \stackrel{\text{def}}{=} \max(0, \max(x,y))$ *and* $\beta(x,y,z) \stackrel{\text{def}}{=} [x > 0 \wedge y > 0]z$.

We can then modify our approximate recursion from before.

**Definition 12.** We define, $\hat{g}_{h,v}^{s,a}(m+1,1,u) \overset{\text{def}}{=} \chi_{\{u \geq v\}}$, $\hat{g}_{h,v}^{s,a}(k, S+1, u) \overset{\text{def}}{=} \hat{g}_{h,v}^{s,a}(k+1, 1, u)$ and for $t \leq S$,

$$\hat{g}_{h,\hat{v}}^{s,a}(k,t,u) \overset{\text{def}}{=} \min_{v_{k,t} \in \mathcal{V}} \alpha \Big( \beta \left( C_h(c_k \mid s,a), P_h(t \mid s,a), \bar{C}_{h+1}^*\left(t, v_{k,t}\right) \right),$$
$$\hat{g}_{h,v}^{s,a}\left(k, t+1, \lfloor u + C_h(c_k \mid s,a)P_h(t \mid s,a)v_{k,t} \rfloor_{\mathcal{G}} \right) \Big). \tag{24}$$

**Approximation.**   Lastly, our rounding now occur error over time, space, and cost. Thus, we simply need to slightly modify our rounding functions. The main change is we use $\delta \overset{\text{def}}{=} \frac{\epsilon}{H(mS+1)+1}$. We also further relax our lower bounds to $\kappa(v) \overset{\text{def}}{=} v - \delta(mS+1)$ and $\kappa \overset{\text{def}}{=} v(1-\delta)^{mS+1}$ respectively. Our running times correspondingly will have $m^3$ terms now.

### F.2   Infinite Discounting

**Approximations.**   Since we focus on approximation algorithms, the infinite discounted case can be immediately handled by using the idea of effective horizon. We can treat the problem as a finite horizon problem where the finite horizon $H$ defined so that $\sum_{h=H}^{\infty} \gamma^{h-1} r_{max} \leq \epsilon'$. By choosing $\epsilon'$ and $\epsilon$ small enough, we can get traditional value approximations. The discounting also ensures the effective horizon $H$ is polynomially sized implying efficient computation. We just need to assume that $0$-cost actions are always available so that the policy can guarantee feasibility after the effective horizon has passed.

**Hardness.**   We also note that all of the standard hardness results concerning deterministic policy computation carries over to the infinite discounting case even if all quantities are stationary.

### F.3   Faster Approximations

We can significantly improve the running time of our FPTAS. The main guarantee is given in Corollary 1. They key step is to modify Algorithm 3 to use the differences instead of the sums. It is easy to see that this is equivalent since,

$$r_h(s,a) + \sum_{s'} P_h(s' \mid s,a)v_{s'} \geq v \iff v - \sum_{s'} P_h(s' \mid s,a)v_{s'} \leq r_h(s,a).$$

Since rounding down the differences make them larger, it becomes harder to be below $r_h(s,a)$. Consequently, we now interpret $\kappa$ as an upper bound for $r_h(s,a)$ instead of a lower bound on $v$ The approximate dynamic programming method based on differences can be seen in Definition 13.

**Definition 13.** Fix a rounding down function $\lfloor \cdot \rfloor_{\mathcal{G}}$ and upper bound function $\kappa$. For any $h \in [H]$, $s \in \mathcal{S}$, $v \in \mathcal{V}$, and $u \in \mathbb{R}$, we define, $\hat{g}_{h,v}^{s,a}(S+1, u) = \chi_{\{u \leq \kappa(r_h(s,a))\}}$ and for $t \leq S$,

$$\hat{g}_h^{s,a}(t,u) \overset{\text{def}}{=} \min_{v_t \in \mathcal{V}} \alpha \left( \beta \left( P_h(t \mid s,a), \bar{C}_{h+1}^*\left(t, v_t\right) \right), \hat{g}_h^{s,a}(t+1, \lfloor u - P_h(t \mid s,a)v_t \rfloor_{\mathcal{G}}) \right). \quad \text{(DIF)}$$

The recursion is nearly identical to the originally, and unsurprisingly, it retains the same theoretical guarantees but in the reverse order. The guarantees can be seen in Lemma 13, which is straightforward to prove following the approach in the proof of Lemma 1.

**Lemma 13.** *For any $t \in [S+1]$ and $u \in \mathbb{R}$, we have that,*

$$\hat{g}_h^{s,a}(t,u) = \min_{\mathbf{v} \in \mathcal{V}^{S-t+1}} \hat{g}_{h,\hat{v}}^{s,a}(t)$$
$$s.t. \quad \tilde{\sigma}_{h,\mathbf{v}}^{s,a}(t,u) \leq \kappa(r_h(s,a)), \tag{25}$$

*where* $\tilde{\sigma}_{h,\mathbf{v}}^{s,a}(t,u) \overset{\text{def}}{=} \lfloor \lfloor u - P_h(t \mid s,a)v_t \rfloor_{\mathcal{G}} - \ldots - P_h(S \mid s,a)v_S \rfloor_{\mathcal{G}}$.

The difference version so far does not help us get faster algorithms. The key is in how we use it. Since the base case of the recursion is $r_h(s,a)$ and not $v$, we can compute the approximate bellman update for all $v$'s simultaneously. This ends up saving us a factor of $|\mathcal{V}|$ that we had in the original Algorithm 4. The new algorithm is defined in Algorithm 6. The inputs to the recursion are define in Definition 14.

---

**Algorithm 6** Approx Solve

---

**Input:** $(\bar{M}, \bar{C})$

1: $\hat{C}^*_{H+1}(s, v) \leftarrow \chi_{\{v \le 0\}}$ for all $(s, v) \in \bar{S}$
2: **for** $h \leftarrow H$ down to 1 **do**
3:     **for** $s \in \mathcal{S}$ **do**
4:         **for** $a \in \mathcal{A}$ **do**
5:             $\hat{g}^{s,a}_h(S+1, u) \leftarrow \chi_{\{u \le \kappa(r_h(s,a))\}} \; \forall u \in \hat{\mathcal{U}}^{s,a}_h$
6:             **for** $t \leftarrow S$ down to 1 **do**
7:                 **for** $u \in \hat{\mathcal{U}}^{s,a}_h$ **do**
8:                     $\hat{v}_{t,a}\hat{g}^{s,a}_h(t, u) \leftarrow$ (DIF)
9:         **for** $v \in \mathcal{V}$ **do**
10:            $a^*, \hat{C}^*_h(s, v) \leftarrow \min_{a \in \mathcal{A}} c_h(s, a) + \hat{g}^{s,a}_h(1, v)$
11:            $\pi_h(s, v) \leftarrow a^*$
12: **return** $\pi$ and $\hat{C}^*$

---

**Definition 14.** For any $h \in [H]$, $s \in \mathcal{S}$, and $a \in \mathcal{A}$, we define $\hat{\mathcal{U}}^{s,a}_h(1) \overset{\text{def}}{=} \mathcal{V}$ and for any $t \in [S]$,

$$\hat{\mathcal{U}}^{s,a}_h(t+1) \overset{\text{def}}{=} \bigcup_{v_t \in \mathcal{V}} \bigcup_{\hat{\sigma} \in \hat{\mathcal{U}}^{s,a}_h(t)} \left\{ \lfloor \hat{\sigma} - P_h(t \mid s, a)v_t \rfloor_{\mathcal{G}} \right\}. \tag{26}$$

**Proposition 6.** *The running time of Algorithm 6 is $O(HS^2 A|\mathcal{V}|\hat{\sigma})$.*

**Corollary 1** (Running Time Improvements). *Using Algorithm 6, the running time of our additive-FPTAS becomes $O(H^5 S^4 A r^2_{max}/\epsilon^2)$, and the running time of our relative-FPTAS becomes $O(H^5 S^4 A \log(\frac{r_{max}}{r_{min}p_{min}})^2/\epsilon^2)$*

**Approximation Details.** Although the running times our clear from removing the factor of $|\hat{V}|$, we need to slightly alter our approximation schemes for this to work. First, we need to use $\kappa(r_h(s, a)) \overset{\text{def}}{=} r_h(s, a) + \delta$ for the additive approximation. The proof from before goes through almost identically.

However, for the relative approximation, no choice of upper bound can ensure enough feasibility. Thus, we simply use $\kappa(r_h(s, a)) \overset{\text{def}}{=} r_h(s, a)$ and apply a different analysis. We also note that technically, differences can become negative. To deal with this the relative rounding function should simply send any negative number to 0: $\lfloor -x \rfloor_{\mathcal{G}} \overset{\text{def}}{=} 0$. The analysis is mostly the same, but the feasibility statement must be slightly modified.

**Lemma 14.** *Suppose all rewards are non-negative. For any $h \in [H + 1]$ and $(s, v) \in \bar{S}$, $\hat{C}^*_h(s, \lfloor v(1 - \delta)^{H-h+1} \rfloor_{\mathcal{G}}) \le \bar{C}^*_h(s, v)$.*

The idea is that since no fixed upper bound can capture arbitrary input values, we simply input relative values. Then, the feasibility part of Lemma 11 goes through as before. The proofs mostly remain the same, but the rounding must again change. We must now start at the smaller $v_{min}$ that is the original $v_{min}$ scaled by a factor of $(1 - \delta)^H$ to ensure that $\lfloor V^*_M(1 - \delta)^H \rfloor$ is in $\hat{V}$. This makes $\hat{V}$ larger, but not by too much as we argued in previous analyses.

