# OpenReview forum: "Deterministic Policies for Constrained Reinforcement Learning in Polynomial Time"
_NeurIPS.cc/2024/Conference — NeurIPS 2024 poster_

### Official Review · Reviewer_SLVm · 2024-06-30

**Soundness:** 2
**Presentation:** 1
**Contribution:** 1
**Rating:** 4
**Confidence:** 4

**Summary:**

The authors propose an algorithm that efficiently computes near-optimal deterministic policies for constrained reinforcement learning (CRL) problems. The proposed algorithm comprises three ideas: 1) value-demand augmentation, 2) action-space approximate dynamic programming, and 3) time-space rounding. Under some assumptions, their algorithm constitutes a fully polynomial-time approximation scheme (FPTAS) for a few cost criteria. This class requires that the cost of a policy can be computed recursively over both time and (state) space, which includes classical expectation, almost sure, and anytime constraints.

**Strengths:**

**Novelty.** I think this paper brings new insights into the field of safe RL (based on constraints) or CMDPs. I think the authors provide unique views for addressing or solving constrained RL problems.

**Cover MDP.** It is an interesting approach to formulate a CMDP that covers diverse formulations for constrained RL. There have similar attempts as I will mention in Weakness, the proposed one is novel as far as I know.

**Algorithm.** I think the proposed algorithm is technically sound. It is a good property to solve diverse problems using a single algorithm in a reasonable computational time.

**Weaknesses:**

**Unclear difference from previous work.** I do not think this paper clearly conveys the differences or contributions compared to previous work. I think this paper is not written after fully surveying existing work. Within the several major claims the authors make, there are several existing works where similar attempts were made. For example, in safe RL literature, there have been several papers on state augmentation or generalized formulations:
- Sootla, Aivar, et al. "Sauté RL: Almost surely safe reinforcement learning using state augmentation." ICML, 2022.
- Wachi, Akifumi, Xun Shen, and Yanan Sui. "A Survey of Constraint Formulations in Safe Reinforcement Learning." IJCAI (2024).

I know existing works have not made the same contributions as this paper, but it is significant to properly make the differences clear rather than exaggerate the contributions.

**Theoretical results.** Though I think the theoretical results are good, I think it is better to compare the theoretical results with existing ones. I understand no existing algorithm can solve diverse safe RL problems, unlike the proposed ones. However, there are a bunch of theoretical results on CMDPs based on expectations (and recent papers derive the upper bounds for other problem formulations). It is necessary to compare the upper bound or strength of the assumptions between the SOTA algorithm and the proposed ones.

**Experiments.** I highly respect theoretical ML papers, but I think this paper needs empirical evaluation because the main claim is efficient computation. The algorithm is complicated and it is questionable whether the proposed algorithm actually works well even in toy problems.

In summary, this paper presents novel and interesting insights, but this paper is not written based on existing research. Thus, the contribution is not clear and I feel the contributions of this paper are exaggerated. Also, the theoretical and empirical advantages are not clear, which should be clearly written or supported by additional work. Therefore, I do not think this paper is ready for publication and resubmit to another venue.

**Questions:**

[Q] Could you discuss the computational complexity of the proposed algorithm while referring to those in SOTA algorithms (for constrained RL problems based on expectation)?


### Minor Comments
- Line 117-118: (SR) $\rightarrow$ (TSR)

**Limitations:**

The authors have slightly addressed the limitations. There is no discussion regarding potential negative societal impact of their work.

---

> ### Author Rebuttal · Authors · 2024-08-06
>
> Thank you for your comments! We address the concerns about the contributions of our paper. Our paper is highly focused, addressing only the question proposed on line 29 of our intro: “Can near-optimal deterministic policies for constrained reinforcement learning problems be computed in polynomial time?” Since providing an FPTAS does answer this question, we believe our claims are not exaggerated. We use the term "efficient" as a synonym for polynomial time, which is a convention in computer science, especially in theory works; given this convention, it is accurate to say an FPTAS is efficient. Being a theory work, we did not claim that our algorithms would work well in practice; the only goal of this paper is to derive new theoretical insights as a first step to devising practical algorithms in the future, as stated throughout the paper [21, 29, 58, 316]. Furthermore, we emphasize our results answer three open complexity questions: we prove polynomial-time approximability for 1) anytime-constrained MDPs, 2) almost-sure-constrained MDPs, and 3) deterministic policies for expectation-constrained MDPs, which has been open for nearly 25 years.
>
> For the concerns of past work, we compared many past methods in detail in the marked prior work paragraph [41], where we pointed out each method's time complexity, whether they produced infeasible policies, and whether the policies were suboptimal.
> We emphasize our work studies the computational complexity of finding deterministic policies for a large class of constraints. Importantly, as stated on line 31 of our intro, it is well known that the computation of stochastic expectation-constrained policies is in P, whereas our problem is known to be NP-hard [Feinberg, McMahan]. Consequently, stochastic policy works address a problem in a completely different complexity class, so they cannot be fairly compared to our setting. Similarly, works on learning complexity, including both papers you pointed out, do not apply to our work on computational complexity, and generally must allow constraint violation for learning, unlike our setting where feasibility is critical. Instead, we discussed the papers by McMahan and Castellano, which cover the same constraints as those papers, but from the computational complexity perspective, which is related to our work. Given the space limitations, we tried to cite only the most relevant works concerning the computational complexity of our problem. We were unable to include other interesting papers on deterministic policies, notably Feinberg’s full line of work, that we plan to include in the final paper.
>
> References:
>
> Castellano, Agustin, et al. "Reinforcement learning with almost sure constraints." Learning for Dynamics and Control Conference. PMLR, 2022
>
> Feinberg, Eugene A. “Constrained Discounted Markov Decision Processes and Hamiltonian Cycles.” Mathematics of Operations Research, vol. 25, no. 1, 2000
>
> McMahan, Jeremy, and Xiaojin Zhu. "Anytime-Constrained Reinforcement Learning." International Conference on Artificial Intelligence and Statistics. PMLR, 2024

---

> ### Comment · Reviewer_SLVm · 2024-08-08
>
> Thank you for clarifications! I have read other reviews and authors' rebuttals.
> While I have a deeper understanding of the contribution of this paper, I still feel that existing safe RL papers have been largely ignored. Though this paper is submitted to NeurIPS, this paper has literally no reference paper on safe RL published in NeurIPS/ICML/ICLR. I think this paper is hard to follow and its contribution is unclear for the NeurIPS community. Since many safe RL papers have been published in NeurIPS or relevant conferences, it is better to write this paper based on them for clearly present the contributions of this paper.
>
> I will not be upset if this paper gets accepted, but I am not willing to recommend acceptance.

---

> ### Author Response · Authors · 2024-08-09
> **Follow-up to Reviewer's Comment**
>
> Thank you for your response! We completely agree that we did not reference many Safe RL papers; however, we remind the reviewer that our paper concerns the topic of Constrained RL. Specifically, our paper focuses on the computational complexity of planning for Constrained RL, although our results can technically extend to learning settings. For concrete applications of our work, our method is a polynomial time algorithm for the autonomous vehicle routing problem posed by Hong, and the anytime constraints problem posed by McMahan.
>
> Although Safe RL is a common application of Constrained RL, our work is not directly a Safe RL work. Safe RL primarily focuses on the issue of safe exploration, which does not even appear in the planning setting. We remind the reviewer that the planning problem for most Safe RL formulations lies in P, unlike our NP-hard problem. Given this context, we ask the reviewer: would it be relevant to reference many more Safe RL works other than to show potential applications of our work?

---

> > ### Comment · Reviewer_SLVm · 2024-08-11
> >
> > I think the authors misunderstand the terminology of safe RL. The following eminent survey paper on safe RL (citations > 1900) categorizes these optimization criteria into four groups: (i) the worst-case criterion, (ii) the risk-sensitive criterion, (iii) the constrained criterion, and (iv) other optimization criteria.
> > - Garcıa, Javier, and Fernando Fernández. "A comprehensive survey on safe reinforcement learning." Journal of Machine Learning Research 16.1 (2015): 1437-1480.
> >
> > This paper was published in JMLR in 2015, and the definitions of "safe RL" or "constrained RL" are widely used in the RL community in NeurIPS/ICML. Therefore, I respectfully disagree with the following authors' claim
> > > "Although Safe RL is a common application of Constrained RL, our work is not directly a Safe RL work."
> >
> > Also, the following statement is not true. Many constrained (deep) RL algorithms (e.g., CPO or TRPO-Lagrangian) do not focus on the safe exploration problem. They aim to achieve constraint satisfaction after convergence.
> > > Safe RL primarily focuses on the issue of safe exploration, which does not even appear in the planning setting.
> > - Achiam, Joshua, et al. "Constrained policy optimization." International conference on machine learning. PMLR, 2017.
> >
> > Even if we focus on the constrained RL or constrained MDP, it is unreasonable not to mention any paper on safe RL or constrained RL that has been published in NeurIPS (or relevant conferences).
> >
> > The critical issue is that this paper seems to be written without adequate surveying, which makes the contributions extremely unclear for the NeurIPS community.

---

> > > ### Author Response · Authors · 2024-08-11
> > > **Follow-up to Reviewer's Second Comment**
> > >
> > > We appreciate your feedback! We completely agree that some safe RL papers study convergence; we in no way intended to make a sweeping statement about all safe RL papers and we apologize for the imprecise phrasing of our statement. As for the comment on constrained RL, we mention that some constrained RL papers, such as those addressing constrained resource problems, do not concern safety. Similarly, some safe RL papers study safe behavior not defined by cumulative cost constraints. Thus, we believe it is fair to say our constrained RL paper is not directly a safe RL paper, but does have applications to safe RL, since constraints may or may not concern safety depending on their intended applications.
> > >
> > > We agree we could cite more works in the paper (and plan to), but we believe we have cited the most relevant ones to our specific problem setting. We note works concerning our problem have appeared in good theory and ML conferences including AISTATS, AAAI, and SODA. In terms of contributions, as we stated in our general rebuttal: "our general framework provides answers to not just one but three open complexity questions spanning two longstanding lines of work: we prove polynomial-time approximability for 1) anytime-constrained MDPs, 2) almost-sure-constrained MDPs, and 3) deterministic policies for expectation-constrained MDPs, which has been open for nearly 25 years [Feinberg]." These specific problems have many interesting applications, which are discussed in the first paragraph of our paper.

---

> ### Comment · Reviewer_SLVm · 2024-08-12
>
> I don't feel there is any progress in the discussion. I think it is simply because the authors have not fully surveyed constrained RL that the authors think that "we believe we have cited the most relevant ones to our specific problem setting".
>
> Because the remaining discussion period is short, I will ask a short and clear question.
>
> **Question.** Could you provide a plan on how to revise related work sections? Please explain what kinds of discussion the authors to planning to add. Here, please list the papers on constrained RL to be cited and discuss the relations with this paper.
>
> Note that I am no longer talking about safe RL. I now focus on constrained RL literature that has been particularly published in NeurIP or ICML.

---

> > ### Comment · Area_Chair_FgZ1 · 2024-08-12
> >
> > Dear reviewer SLVm,
> >
> > Your point is well taken, that the authors are encouraged to make more connections to adjacent areas that receive more attention in the NeurIPS/ICML communities. I do think the authors have agreed to this sentiment in general, and I am inclined to leaving the detailed execution to the authors.
> >
> > After all, this is a planning / computational complexity paper. I understand that such a topic may feel a bit distant to many in the community nowadays, but such areas (with overlap with planning venues like ICAPS and TCS venues like SODA/STOC) have always been within the scope of NeurIPS and part of theoretical foundation of RL. I welcome specific and concrete citation suggestions that are directly relevant to the paper from reviewers with close expertise on the topic. For a computational complexity paper, such citations would be prior papers that establish upper / lower bounds on problems closely related to the one studied in the current work. Otherwise I am inclined to treat additional citations as "soft suggestions" and the authors can use their judgement to include whatever they feel relevant.
> >
> > AC

---

> > > ### Comment · Reviewer_SLVm · 2024-08-12
> > >
> > > Dear AC FgZ1,
> > >
> > > If the AC considers that this paper is written so that the contributions are clearly explained for the NeurIPS community, I respect the decision.

---

### Official Review · Reviewer_DPnW · 2024-07-10

**Soundness:** 4
**Presentation:** 3
**Contribution:** 3
**Rating:** 5
**Confidence:** 3

**Summary:**

The paper proposes a novel algorithm for constraint MDPs whose constraints have TR properties. The algorithm converts the original problem to an unconstraint MDP. To address the challenge of high complexity of the proposed approach, the method discovers that for constraints with TSR properties, it can accelerate the Bellman update. Furthermore, the paper proposes the approximate version of the problem by rounding the values effectively.

**Strengths:**

1. The analysis is comprehensive and the logic is strong.
2. The fast bellman update for TSR constraints and the approximate rounding are interesting but very useful in practical computation.
3. The overall writing is good, esp. for math notations and explanation.

**Weaknesses:**

1. The problem extends knapsack problems to constraint MDP. It is unclear if this problem is novel or is a common interest in the RL and control community.
2. It is hard to think of practical problems that fit the setting on TSR, could you provide examples that needs constraint MDP with TSR (TR) constraints while cannot be modeled as a single MDP problem with cleverly designed rewards?
3. The overall algorithm and the improvement are important but also seems like straight-forward extensions on the basic algorithms. The overall complexity is still very high at last.

**Questions:**

1. Could you provide examples that needs constraint MDP with TSR (TR) constraints while cannot be modeled as a single MDP problem with cleverly designed rewards?
2. Is there existing works  using the same problem definition?

**Limitations:**

1. The complexity is still high.

---

> ### Author Rebuttal · Authors · 2024-08-06
>
> Thank you for your comments! Although we introduce TSR constraints, they generalize the main constraints studied by the constrained reinforcement learning (CRL) literature, including expectation, almost sure, and anytime constraints. CRL is an entire subarea of RL that was born as early as 1981 [Kallenberg] and is often used by the safe RL community. Since CRL has hundreds of impressive works, we cannot easily convey the many motivations for constrained problems. Instead, we give an example of anytime constraints: self-driving cars must not run out of fuel. Here, it is easy to see that reward shaping could only work if we include current fuel levels in the state space, but this would take exponential time [McMahan], which necessitates the constraint formulation of the problem. McMahan and Castellano point out many more applications of anytime and almost-sure constraints. Finding deterministic policies under expectation constraints was first motivated by Feinberg and Schwartz and has been a focus of works on autonomous vehicles [Hong, Geißer]. Many other applications, including medicine, can be found in the first paragraph of our paper.
>
> We note the TSR constraints, and even the constraints they generalize, have a very different mathematical structure from the knapsack problem. Adding stochasticity to the knapsack problem increases the complexity due to the non-trivial adaptivity gap, which forces optimal solutions to take the form of adaptive “policies” rather than a simple set of knapsack items [Dean]. Introducing MDPs only further increases the complexity [McMahan]. Consequently, our problem is non-trivial compared to the knapsack problem, and our methods are not straightforward extensions of knapsack algorithms. Moreover, our algorithm is the first ever provable approximation algorithm for almost sure and anytime constraints, let alone an FPTAS, which is known to be the best approximation guarantee possible. Furthermore, our algorithm is the first ever FPTAS for expectation constraints for general MDPs since the NP-hardness result from nearly 25 years ago [Feinberg]. Although the complexity is arguably high, our method is still the first provable approximation algorithm for these problems and gives insights into breaking through the exponential time barrier. We hope these insights will lead to even faster algorithms in future work.
>
> References:
>
> Dean, Brian C., Michel X. Goemans, and Jan Vondrák. "Approximating the stochastic knapsack problem: The benefit of adaptivity." Mathematics of Operations Research 33.4 (2008): 945-964
>
> Feinberg, Eugene A. “Constrained Discounted Markov Decision Processes and Hamiltonian Cycles.” Mathematics of Operations Research, vol. 25, no. 1, 2000
>
> Geißer, Florian, et al. "Optimal and heuristic approaches for constrained flight planning under weather uncertainty." Proceedings of the International Conference on Automated Planning and Scheduling. Vol. 30. 2020.
>
> Hong, Sungkweon, et al. "An anytime algorithm for chance constrained stochastic shortest path problems and its application to aircraft routing." 2021 IEEE International Conference on Robotics and Automation (ICRA). IEEE, 2021
>
> Kallenberg, L. C. M. "Unconstrained and constrained dynamic programming over a finite horizon." 1981.
>
> McMahan, Jeremy, and Xiaojin Zhu. "Anytime-Constrained Reinforcement Learning." International Conference on Artificial Intelligence and Statistics. PMLR, 2024

---

### Official Review · Reviewer_iqFB · 2024-07-13

**Soundness:** 2
**Presentation:** 2
**Contribution:** 3
**Rating:** 5
**Confidence:** 2

**Summary:**

The paper presents an algorithm that computes deterministic policy for constraint RL. The work is majorly theoretical. The proposed algorithm for the worst-case analysis achieves the best approximation guarantees. Their approach incorporates ideas from state augmentation with value function, approximate dynamic programming, and time-space rounding.

**Strengths:**

- Computes a near-optimal policy
- Consider a diverse range of constraint functions
- Algorithm belongs to FPTAS

**Weaknesses:**

- Motivation sounds weak or rather superficial. In the case of stochastic policies, why can one not apply the mean of the policy to get a deterministic behaviour?

- All the Algorithms are difficult to parse/not self-contained. What does it mean if a definition takes an input of M, C?  Maybe I missed it somewhere, what is V^s? Would be good if you can define the symbols in the algorithm at least input.

- Theoretical statement doesn't contain assumptions. It would be good if you could incorporate assumptions in the theoretical statements, especially in propositions and theorems.

- Finite state-action space, perhaps rather difficult to then have meaningful applications among the ones that the authors mention in the first paragraph, e.g., autonomous driving. On a high level I am wondering, since you are also solving for a finite horizon case, why would one not apply an open loop MPC type approach for a constraint finite horizon application (that also results in deterministic policies and is polynomial time for finite H).

**Questions:**

- Why is Proposition 2 a proposition but not just an example?
- remove line 144. It creates more confusion than helps to understand. E.g., use of not defined terminology of covering MDP now and value augmentation.
- In line 148, what does "this covering program" mean? Does it correspond to the reversed objective?
- why value-demand augmentation and not value-function augmentation?

---

> ### Author Rebuttal · Authors · 2024-08-06
>
> Thank you for your comments!
>
> We first address the weakness comments:
> 1. We do not believe our motivation is weak as it comes from two longstanding lines of work: (1) deterministic policies, which are critical for multi-agent coordination and for autonomous vehicles [Hong, Geißer], and (2) Anytime/Almost-Sure constraints which are prevalent in any task with resource consumption and any safety critical system [McMahan, Castellano]. More applications, including medicine, can be found in our first paragraph. The fact that simple derandomization strategies fail is the topic of an entire line of work that started with Feinberg in 2000 and culminated with our paper. It is difficult to fully capture the intuition from this line of work, but one way to see that any simple derandomization approach fails is to note that stochastic policy computation is in P, but deterministic policy computation is NP-hard as mentioned on line 31. As such, it is immediate that no easy derandomization strategy can exist unless P = NP. We emphasize again that we do not only solve the deterministic policy problem but also solve the anytime and almost-sure-constrained problems, which is a major contribution of our paper. The fact that we can solve both of these problems with one framwork is a major strength of our work.
> 2. The definition of the covering MDP depends on M and C. Consequently, we used Definition(M,C) to indicate that this particular M and C should be used to construct the covering MDP; we will clarify this in the writing. $v_s$ is defined informally on line 165, formally on line 168, and compactly in definition 2; note we use standard notation for indexing a vector. To our knowledge, all quantities are defined according to standard theory convention; if there are any undefined symbols, we are happy to fix them.
> 3. Perhaps surprisingly, the only assumption we need in nearly all of the paper is that the cost function is TSR. The only other assumptions are the reward assumptions that are explicitly stated in the two approximation theorems, Theorem 2 and 3. Thus, all theoretical statements do contain their assumptions. We ask the reviewer: what assumptions do you feel have not been included?
> 4. We completely agree it would be nice to extend beyond the tabular case in future work. Since not even the tabular setting has been resolved until now, our work serves as an important first step to devising more practical algorithms in the future.
> 5. Since past approaches have been unable to solve this problem til now, we attempted a different approach inspired by the approximation algorithm community to achieve provable guarantees.
>
> To your questions:
> 1. They are not examples because they need non-trivial mathematical proof to verify they satisfy the TSR condition. Anytime constraints, especially, were non-trivial to massage into the TSR form. The proofs are given in the appendix.
> 2. Yes, the covering program corresponds to flipping the objective and the constraint similarly to duality. Packing and covering programs are common in optimization, especially combinatorial optimization, so we used those terms to connect to that literature.
> 3. The value demand is a total value goal the agent must hit; it is different from the actual value it has accumulated. We use the term demand just as in general optimization and network flow theory to indicate an amount requirement on a resource.
>
> References:
>
> Feinberg, Eugene A. “Constrained Discounted Markov Decision Processes and Hamiltonian Cycles.” Mathematics of Operations Research, vol. 25, no. 1, 2000
>
> Geißer, Florian, et al. "Optimal and heuristic approaches for constrained flight planning under weather uncertainty." Proceedings of the International Conference on Automated Planning and Scheduling. Vol. 30. 2020
>
> Hong, Sungkweon, et al. "An anytime algorithm for chance constrained stochastic shortest path problems and its application to aircraft routing." 2021 IEEE International Conference on Robotics and Automation (ICRA). IEEE, 2021
>
> McMahan, Jeremy, and Xiaojin Zhu. "Anytime-Constrained Reinforcement Learning." International Conference on Artificial Intelligence and Statistics. PMLR, 2024

---

> ### Comment · Area_Chair_FgZ1 · 2024-08-11
> **"mean of a policy"**
>
> Dear reviewer iqFB,
>
> > why can one not apply the mean of the policy to get a deterministic behaviour?
>
> Note that you cannot do that in MDPs in general. In standard MDP formulations, actions are categorical variables, so you cannot simply "take the mean". Your comment would make sense in continuous control problems where actions are vectors in Euclidean spaces, but that requires additional structural assumptions on the dynamics (e.g., this might be true in linear systems).

---

### Official Review · Reviewer_pADD · 2024-07-16

**Soundness:** 4
**Presentation:** 4
**Contribution:** 4
**Rating:** 8
**Confidence:** 2

**Summary:**

The paper provide a novel polynomial-time algorithm for finding near-optimal deterministic policies in constrained MDPs with a large family constraints beyond expectation. The algorithm first considers in the dual covering problem minimizing cost for constrained value. The new problem is then viewed as solving a new augmented MDP with "value demands." This MDP is more prone to approximate algorithms via approximate dynamic programming and careful rounding. In the process, the class of constraints supported by the algorithm (TSR constraints) is a step towards the class of constraints that allow polynomial-time algorithms.

**Strengths:**

- The paper solves a significant problem in the area. Apart form the new tractability result, the TSR class for constraints includes most kinds of constraints of interest and only excludes the ones that are known to be impossible to solve. This is a step towards classification of constraints that admit FPTAS.
- I am not an expert in the area, and I cannot comment on novelty of techniques used by the paper in confidence. However, I find them not only novel, but also potentially inspiring for RL theory.
- The paper is well-written and easy to follow despite its technical complexity.

**Weaknesses:**

I think more discussions on the optimality/tightness of each step of algorithm design would benefit the paper. I suppose some of the operations/steps are specific to this problem and therefore investigation of the optimality the whole algorithm might require a lot effort possibly in another paper. However, I wonder if for some of the steps we can have insights on whether the operation is optimal or not. For example, if I am not wrong, switching to the covering problem was harmless, same for the reduction to the value augmented MDP. I am not sure how optimal handling the augmented MDP is. However, I guess due to the MDP formulation some existing results may apply. Perhaps the authors can provide some insights on each of the steps.

Minor comments:
-  Some small comments in the paper for quantities that will be discussed later would be welcome. Specifically, in the discussions on tractability of solving the cover MDP the lack of knowledge of the set $V$ is not brought up. Only later in the paper this issue is addressed.

- The Objective of the cover MDP could be better clarified. The objective appears to be determined by the function $f$ as opposed to the conventional summation and expectations of value functions.

**Questions:**

Please see the weaknesses. Also I wonder whether the proof techniques to focus on covering problem in cMDPs, and approximate dynamic programming are common in the literature. If so, more citations could be useful.

**Limitations:**

The paper does a good job outlining the limitations.

---

> ### Author Rebuttal · Authors · 2024-08-06
>
> Thank you for your comments! The main techniques used in this paper are packing-covering equivalence (essentially, strong duality), dynamic programming, and rounding. These are general techniques found in standard approximation algorithms textbooks such as the book “The Design of Approximation Algorithms” cited in our paper. Moreover, dynamic programming is crucial to most of MDP theory. How we applied these techniques, such as creating a covering MDP and applying dynamic programming to the action space, is unique to this problem. We are unaware of any other paper that uses these general techniques in the same way.
>
> As for the optimality of each step of our reduction, we note no step harms the optimality of the solution except for the rounding steps. In terms of fine-grained time complexity, it is hard to say whether reducing to the covering problem somehow limits how fast algorithms could be. Resolving that question would require a time complexity lower bound for both the original and the covering problem for comparison. Proving such lower bounds would be fairly complicated and is far beyond the scope of this paper, but would be very interesting for future work.

---

> > ### Comment · Reviewer_pADD · 2024-08-13
> > **Rebuttal Acknowledgement**
> >
> > I thank the authors for their rebuttal. I have read all the comments and maintain my score.

---

### Author Rebuttal · Authors · 2024-08-06

Since the reviewers agree with the mathematical correctness of our results, we emphasize the significance of our contributions.
To summarize, our general framework provides answers to not just one but three open complexity questions spanning two longstanding lines of work: we prove polynomial-time approximability for 1) anytime-constrained MDPs, 2) almost-sure-constrained MDPs, and 3) deterministic policies for expectation-constrained MDPs, which has been open for nearly 25 years [Feinberg]. Before our work, no provable approximation algorithms had been known for any of these problems, absent a special case of expectation constraints [Khonji], and we designed algorithms satisfying the conditions of an FPTAS, which is the best possible approximation guarantee.

Computing optimal deterministic policies under expectation constraints was shown to be NP-hard in 2000 [Feinberg]. Since then, only one work, to our knowledge, has successfully derived a non-trivial approximation algorithm, but only for the constant horizon setting [Khonji]. Our paper is the first to fully settle the computational complexity of deterministic policies since the original hardness result nearly 25 years ago. Moreover, we provide an FPTAS, known to be the best possible approximation guarantee.

Since as early as 2011 [Xu], the literature has argued for alternative constraints to handle many real-world applications. Recently, there has been a strong push for anytime [McMahan] and almost-sure constraints [Castellano]. Our work is also the first to provide any provable approximation algorithm for these constraints, let alone an FPTAS. Solving these newer constraints alone is a notable contribution of our work, and our general framework can solve constraints beyond even these.

References:

Castellano, Agustin, et al. "Reinforcement learning with almost sure constraints." Learning for Dynamics and Control Conference. PMLR, 2022.

Feinberg, Eugene A. “Constrained Discounted Markov Decision Processes and Hamiltonian Cycles.” Mathematics of Operations Research, vol. 25, no. 1, 2000

Khonji, Majid, Ashkan Jasour, and Brian C. Williams. "Approximability of Constant-horizon Constrained POMDP." IJCAI. 2019

McMahan, Jeremy, and Xiaojin Zhu. "Anytime-Constrained Reinforcement Learning." International Conference on Artificial Intelligence and Statistics. PMLR, 2024

Xu, Huan, and Shie Mannor. "Probabilistic goal Markov decision processes." IJCAI Proceedings-International Joint Conference on Artificial Intelligence. Vol. 22. No. 3. 2011.

---

### Decision · Program_Chairs · 2024-09-25

**Decision:**

Accept (poster)

**Comment:**

The paper presents novel theoretical results in finding near-optimal deterministic policies in constrained RL. The reviewers generally appreciate the theoretical achievement and the relevance of the problem. There are concerns that the work is leaning heavily towards computational complexity and planning communities, and the authors are encouraged to make more connections to closely related topics in the NeurIPS community.